# TOMVALLEY: EVALUATING THE THEORY OF MIND REASONING OF LLMS IN REALISTIC SOCIAL CONTEXT

## ABSTRACT

As large language models (LLMs) are increasingly involved in human society, some studies try to evaluate LLMs' capability of theory of mind (ToM), which is about the understanding and reasoning of others' mental states and possible actions. However, these previous works simplify the ToM capability required in real social contexts during their evaluations. This can be reflected in three aspects: (1) most evaluations focus on a **static mental state** after several social scenarios while ignoring the changes of mental states across different scenarios; (2) they mainly consider **independent mental states**, however different kinds of mental states (beliefs, intentions, and emotions) and actions can influence one another in our real life; (3) there is an **absence of social settings and character profiles** in their evaluation, even though humans can effortlessly obtain and utilize this information in ToM reasoning processes. This lack can underestimate the abilities of LLMs. This paper aims to evaluate LLMs' ToM capability in closer alignment with a realistic social context. Correspondingly, we propose a new benchmark, named TOMVALLEY, which alleviates the limitations mentioned above of previous works. Specifically, the benchmark is constructed using a framework that includes four steps: social background determination, mental state sketch, social scenario design, and rule-based question generation. Overall, there are 1100 social contexts and 78100 questions about characters' mental states. The quality of the benchmark is manually verified. Additionally, we evaluate ten popular LLMs on TOMVALLEY. Experimental results suggest that LLMs' performances are significantly inferior to human levels by 11%. Subsequent investigation indicates that LLMs are ineffective at interpreting alterations in mental states across social scenarios. Furthermore, we observe that LLMs are incapable of addressing compositional questions that necessitate multi-hop reasoning within the social context.[1]

## 1 INTRODUCTION

Theory of Mind (ToM) refers to the capacity to understand and reason about the mental states of others (e.g., beliefs, intentions, and emotions) and predict their next actions by inferring their mental states (Leslie et al., 2004; Call & Tomasello, 2008; Apperly & Butterfill, 2009). For human beings, ToM is a foundational capability in various daily social interactions (Turner, 1988), such as maintaining relationships (Hughes & Leekam, 2004), making decisions (Carlson & Moses, 2001), and enhancing peer popularity (Slaughter et al., 2015). As LLMs develop, they become increasingly involved in social activities and interact with humans daily. In these interactions, LLMs' ToM capabilities significantly impact their performance. For instance, LLMs are expected to understand and reason about others' beliefs, emotions and intentions, and even influence the users' actions in support conversations (Liu et al., 2024c; Wang et al., 2024a).

Realizing the importance of LLMs' ToM reasoning capacity, previous studies have proposed various benchmarks to evaluate LLMs, such as SocialIQA (Sap et al., 2019), BigToM (Gandhi et al., 2024), and TOMBENCH (Chen et al., 2024). Typically, these benchmarks contain questions about

---

[1] https://anonymous.4open.science/r/ToMValley-ICLR/README.md

a character's mental state, such as "given [the social scenario(s)], what does [a person] believe?" However, these works simplify the ToM capability required in real social contexts. The simplification is mainly reflected in the following three aspects. (1) **Static mental states**. Questions in most benchmarks only require LLMs to deduce the character's final mental state after one or several social scenarios. However, the mental state can change across different scenarios in our real-life social context. (2) **Independent mental states**. Most benchmarks only test one kind of mental state in one scenario. However, there exists an inter-relationship among different kinds of mental states (e.g., beliefs, intentions, and emotions) (D'Andrade, 1995; Wellman, 1990). For instance, belief and emotion will influence one's intention. (3) **Absence of social locations and character profiles**. Humans can effortlessly obtain and utilize social location information(e.g., cafes) and some character profiles in their ToM reasoning (Bretherton & Beeghly, 1982; Gönültaş et al., 2020). Their absence can lead to underestimation of LLMs' ability and prohibit us from appropriately determining the causes of LLMs' failure cases: whether it is due to the model's inherently inadequate ToM reasoning abilities or the lack of corresponding information.

This paper aims to evaluate LLMs' ToM capability in closer alignment with real-world social contexts. Correspondingly, we construct a novel benchmark, named **TOMVALLEY**, and try to alleviate the three mentioned limitations of previous works by the following three considerations. (1) **Dynamic mental states**. Rather than merely assessing the final mental state after a social scenario, TOMVALLEY incorporates questions that explore the alterations of mental states across multiple continuous social scenarios, corresponding to the dynamic nature of the mental state. (2) **Intradependent mental states**. TOMVALLEY investigates how different types of mental states influence one another. This involves constructing the intradependent relationships between mental states, shaping the progression of the social scenario plot based on the relationships, and tailoring questions to evaluate LLM's ability to reason how different mental states influence each other. (3) **Provision of social location and character profiles**. Each social context in TOMVALLEY includes detailed social locations and character profiles, offering LLMs rich contextual information to reason about the characters' mental states.

Notably, TOMVALLEY is constructed with the framework shown in Figure 1. As illustrated, it includes four main processes: (1) determining a social background, including the social location, character profiles, and the relationship between characters; (2) sketching the main character's mental states across different social scenarios; (3) designing several social scenarios that happen on the main character based on the sketched mental states and the social background; (4) generating questions, whose answers and options can be directly extracted from the output of step (2). The data in TOMVALLEY include the social background, social scenarios, and the questions (i.e., outputs of step (1), (3), and (4) of the framework). In summary, TOMVALLEY contains 1100 social contexts (2,200 characters and 5,500 social scenarios) and 78100 questions related to mental states. We employ human annotators to evaluate and verify the quality of TOMVALLEY, and establish a human baseline. Feedback from annotators indicates that the social backgrounds and scenarios closely resemble real-life interactions, the questions and respective options are reasonable, and the ground truths are validated despite humans not being 100% correct when establishing the human baseline. In addition, we evaluate ten popular LLMs' ToM capabilities using both vanilla and CoT prompting. Experimental results show that current LLMs underperform humans significantly on the ToM capability: even the best LLM's performance, achieved by GPT-4o, is lower than humans' by 11%. Further analysis shows that LLMs' performance will decrease when useful information is presented in the middle of input. Moreover, we find that LLMs are incapable of processing compositional problems that necessitate multi-hop reasoning within the social context.

Overall, our contributions are as follows: (1) We propose to evaluate LLMs' ToM capabilities in closer alignment with the real-world social contexts with the consideration of dynamic and intradependent mental states as well as the provision of social location and character profiles in reasoning processes. (2) We introduce TOMVALLEY, a benchmark featuring personified individuals, diverse social locations, and dynamic and intradependent mental states. Through human evaluation, we demonstrate the benchmark's uniqueness, difficulty, and high quality. (3) We conduct a thorough assessment of ten popular LLMs, and compare their performances against human performance, and provide an in-depth analysis of their limitations.

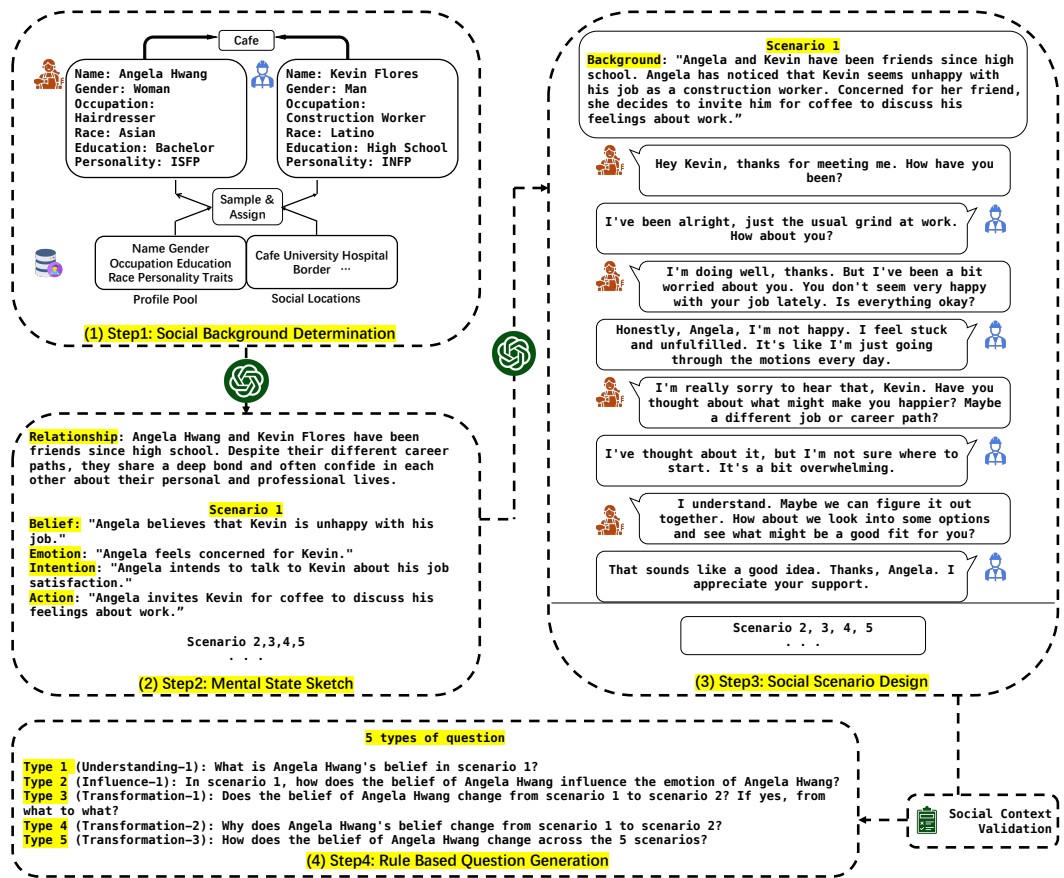

Figure 1: The framework used to generate the TOMVALLEY.

## 2 RELATED WORK

### 2.1 TOM BENCHMARKS

Arguably, infants as young as 12 months of age can attribute mental states to others, demonstrating theory of mind reasoning (Onishi & Baillargeon, 2005). Theory of mind appears to be an innate potential ability in humans that requires social and other experiences over many years for its full development. Researchers have begun to probe whether LLMs possess a Theory of Mind ability comparable to that of humans, as they have reached and occasionally surpassed human performance in some task-solving and reasoning tasks. Nematzadeh et al. (2018); Le et al. (2019); Wu et al. (2023) apply the Sally-Anne Test (Baron-Cohen et al., 1985) and bAbi (Weston et al., 2015) to test LLMs' ToM ability in the aspect of false belief, and they find that LMs' performance is significantly lower than humans. Kosinski (2023); Bubeck et al. (2023); van Duijn et al. (2023) report GPT-series' performance on Sally-Anne Test is comparable to or outperforms children aged 7 and 10. However, Ullman (2023); Shapira et al. (2024); Kim et al. (2023); Sap et al. (2022) propose that LLMs clearly lack of ToM without robust performance and LLMs are prone to shortcuts and spurious correlations. Apart from the test in the aspect of belief, Sap et al. (2019); Xu et al. (2024); Chen et al. (2024) construct benchmarks to test LLMs' ToM ability for emotion, intention, and perception. Previous evaluations suffer from one or more of the following issues: static mental states, independent mental states, lack of the statement of social location, and absence of clear character information. Our work aims to develop a scalable, novel framework and benchmark to understand the ToM reasoning of language models in the dynamic social context.

## 2.2 PROCESS-LEVEL EVALUATION FOR REASONING.

Although it is essential to curate comprehensive and appropriate data for benchmarks, it is equally important to implement rigorous evaluation methodologies that scrutinize the step-by-step reasoning processes of AI models. The primary objective of most current benchmarks is to evaluate the model's output relative to the standard answer at the answer level. Some recent works (Uesato et al., 2022; Lightman et al., 2024; Wang et al., 2024b) have begun to concentrate on the intermediate math reasoning stages of the models. In the social interaction evaluations, Zhou et al. (2024) propose SOTOPIA-EVAL to evaluate the multi-faceted social interactions, which not only require completing major social goals but also multiple implicit goals, such as maintaining relationships, preserving finances, gaining information, keeping secrets, and following social rules. Gandhi et al. (2024) propose a framework for procedurally designing synthetic ToM evaluations from causal templates to interpret the failure cases in ToM reasoning. However, this work only considers the constrained mental states, without considering the dynamic nature of the mental states, and it also ignores the mutual effect between mental states. Our work aims to integrate the benefits of process-level evaluation, creating a novel approach to generating benchmarks to probe the failure mode of LLMs across comprehensive mental states in the realistic social context.

## 3 TOMVALLEY CONSTRUCTION FRAMEWORK

**Definitions and Preliminaries** We would like to define key terms commonly used in this paper first. *Social Locations* refer to the physical settings of social scenarios. This information is important because it can reflect social norms and influence human behavior Farrow et al. (2017). *Social Scenario* denotes the interactions and activities involving characters. In previous works, such as OpenToM Xu et al. (2024), a single scenario may encompass multiple events occurring at different times. However, in our study, each scenario represents one specific social event at a given moment. *Social Context* encompasses the social background, including social locations and character profiles, as well as multiple social scenarios. This study emphasizes evaluating LLMs' ability to reason about dynamic and intradependent mental states, taking into account social locations and character profiles. Accordingly, we propose a novel framework to generate an evaluation benchmark, as shown in Figure 1. The framework consists of four steps: (1) Social Background Determination, (2) Dynamic and Intradependent Mental State Sketch, (3) Social Scenario Design, and (4) Social Context Validation and Rule-Based Question Generation. Notably, this framework is scalable, allowing us to easily adjust the social context and its corresponding questions by modifying the character number, the dialogue turn, the scenario number, and the question number.

**Step 1: Social Background Determination** The social background contains a social location, character profiles, and the characters' relationship. We collect a location pool and randomly sample one location each time. As for the character's profile, we define seven aspects: surname, name, gender, occupation, education, race, and personality traits. For every aspect, we construct a pool of candidates. After determining the character number in the social context, we construct the profile for each character by sampling one item from each of the seven pools. The relationship between characters is generated with LLMs given the characters' profiles.

**Step 2: Dynamic and Intradependent Mental State Sketch** This work focuses on dynamic and intradependent mental states. For "dynamic", there should be several social scenarios, and the mental states in one scenario can be different from those in the last scenario. We set the number of scenarios as five.[2] For "intradependent", multiple kinds of mental states as well as their intradependences are expected. We include three kinds of mental states: the belief, the emotion, and the intention. In addition, we take the action into account in the sketch since its close relationships to these three mental states. For clarity, we refer to any belief, emotion, intention, or action as a "ToM reasoning item." To sketch such mental states, we prompt LLMs by providing the social background determined in Step 1. Notably, we sketch the mental states of one character, the main character, who LLMs primarily perceive and reason about.

---

[2]Due to our utilization of a commercial model, GPT-4-turbo-2024-04, for scenario design, we have limited the number of scenarios to five to reduce costs while maintaining a more authentic social context. Researchers can easily adjust the scenario number in our framework to meet their needs.

| Item | Number |
|------|--------|
| ToM Reasoning Items | 4 |
| Social Locations | 261 |
| Characters | 2200 |
| Scenarios | 5500 |
| Social Contexts | 1100 |
| Questions | 78100 |
| Average Social Context Length | 457.9 |
| Average Questions Length | 77.5 |

Table 1: TOMVALLEY Statistics.

🏠 Social Location  👥 Relationship  🔗 Dynamic mental states
🖼 Profile  ◈ Intradependent mental states  ✎ Questions Num

| | Plot | 🏠 | 🖼 | 👥 | ◈ | 🔗 | ✎ |
|---|---|---|---|---|---|---|---|
| ToMi | ✗ | ✗ | ✗ | ✗ | ✗ | ✗ | 999 |
| SocialIQA | ✗ | ✗ | ✗ | ✗ | ✗ | ✗ | 37588 |
| Hi-ToM | ✗ | ✔ | ✗ | ✗ | ✗ | ✗ | 1200 |
| OpenToM | ✔ | ✗ | ✔ | ✔ | ✗ | ✗ | 2384 |
| BigToM | ✔ | ✔ | ✗ | ✗ | ✗ | ✔ | 600 |
| TOMBENCH | ✔ | ✗ | ✗ | ✗ | ✗ | ✗ | 2860 |
| TOMVALLEY | ✔ | ✔ | ✔ | ✔ | ✔ | ✔ | 78100 |

Table 2: ToM benchmark Comparison.

**Step 3: Social Scenario Design**   We plot the social scenario in the format of dialogues among characters. This is because dialogue is the primary format through which LLMs perceive and interact with humans. Meanwhile, ToM reasoning in dialogues has seldom been investigated in previous works. The social scenarios focus on a single topic and evolve over time. When designing each social scenario, the utterances of the main character in the dialogue are motivated by their corresponding mental states sketched in Step 2. Specifically, we prompt an LLM to generate the dialogues in five scenarios given the social background (Step 1 output) and the main character's mental state sketch (Step 2 output).

**Step 4: Social Context Validation and Rule-Based Question Generation**   The LLMs' outputs are not reliable all the time. Thus, we need to check the quality of the mental state sketch and social scenarios. In specific, there are three principles: (1) Does the mental state sketch consist of each scenario's belief, emotion, intention, and action? (2) Are the numbers of social scenarios in Step 2 and Step 3 as expected? (3) Are the dialogues coherent with the mental state sketch? Based on these three principles, unqualified instances will be removed by regular matching. Then, we generate questions that can evaluate ToM capability using the qualified instances. We aim to explore five types of questions: (1) (Understanding-1) What is the main character's ToM reasoning item in a specific scenario? (2) (Influence-1) In one scenario, how does mental state A influence ToM reasoning item B? (3) (Transformation-1) Does a ToM reasoning item change from scenario A to scenario B? (4) (Transformation-2) What causes a ToM reasoning item change from scenario A to scenario B? (5) (Transformation-3) How does the ToM reasoning item change across all the scenarios? We designed five templates to generate questions based on the five question types. We use these templates to generate 71 questions for each social context. Notably, action is solely instigated by intention and does not directly influence other toM reasoning item (d'Andrade, 1987). Thus, we do not set any influence question for action. The ground truth and misleading options can be extracted directly from the mental state sketch.

More details of each step can be found in Appendix A.

## 4   TOMVALLEY BENCHMARK

### 4.1   STATISTICS

Leveraging the framework proposed in Section 3 and GPT-4-Turbo[3], we construct the benchmark TOMVALLEY. We set the number of characters as two and the number of scenarios as five for each social context. In addition, the number of locations in the location pool is 261. We generate 1,100 social contexts and 78,100 questions. Each social context includes one social location, two character profiles, the character relationship, and dialogues between characters in five different scenarios. And there are 71 questions about each social context. Table 1 shows the statistics. Moreover, we compare TOMVALLEY with previous benchmarks, and comparison results are shown in Table 2. Although half of previous works have provided a plot (things happened on characters) for ToM reasoning, most of them usually ignore character profiles, relationships, and dynamic mental states. And only our work considers the intradependent mental states.

---

[3]We accessed GPT-4-turbo-2024-04-09 through Microsoft Azure OpenAI service in August 2024.

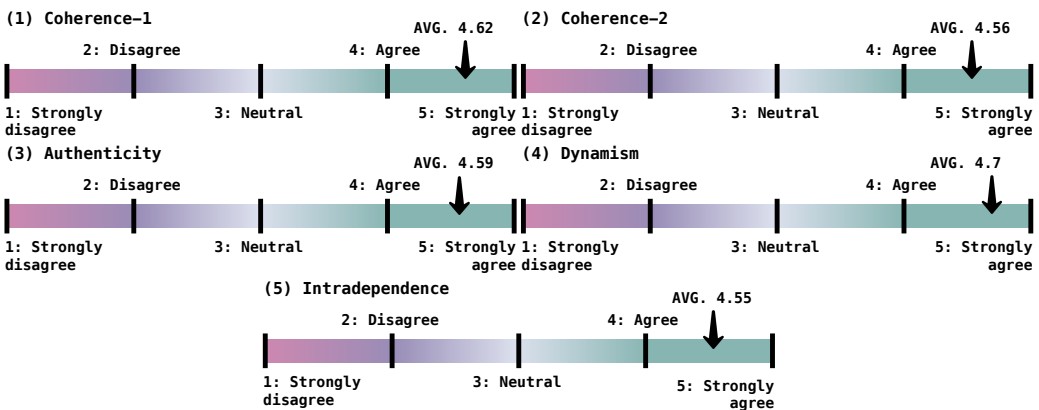

Figure 2: Human evaluation results of the quality of social contexts.

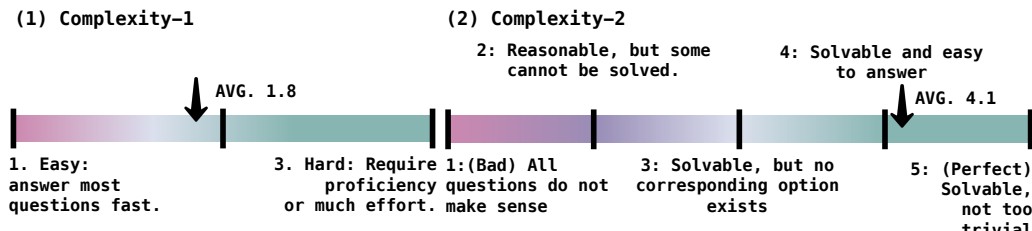

Figure 3: Human evaluation results of the quality of questions.

## 4.2 SOCIAL CONTEXT QUALITY

We evaluate the quality of the social contexts in the benchmark via human evaluation. In specific, we hired five graduate students, and randomly sampled 60 social contexts and 1,420 questions from the benchmark. In addition, we collect the corresponding mental state sketches – outputs of Step 2 in the framework. We present all these contents to the human annotators and ask them to rate the five 5-Likert scale questions: (1) *coherence-1*: to what degree do you agree that the dialogue between characters in the five scenarios is coherent? (2) *coherence-2*: to what degree do you agree that the social background (character profiles, the social location, the character relationship) and the scenarios are coherent? (3) *authenticity*: to what degree do you agree that the social context is authentic and aligns with real life? (4) *dynamism*: to what degree do you agree that the mental state's change of the characters from one scenario to the following scenario is reasonable? (5) *intradependence*: to what degree do you agree that the mental state's influence on one another is reasonable? Here, 1 indicates strongly disagree, while 5 indicates strongly agree.

Figure 2 presents the human evaluation results. For coherence-1 and coherence-2, the average ratings reach 4.62 and 4.56, respectively, indicating the good quality of the social context. The average rating of authenticity reaches 4.59, and 90% of annotators rate with 4 or 5, showing that most annotators agree that social contexts are authentic and align with real life. 93% of annotators agree that the mental state's change of the characters from one scenario to the following scenario is reasonable, and the average rating of dynamism is 4.7.

## 4.3 QUESTION QUALITY AND COMPLEXITY

We would like to explore the quality and complexity of questions in TOMVALLEY: Do the questions have varying degrees of complexity? Correspondingly, we conduct a human evaluation. We invited five human annotators to answer questions about social contexts. Each annotator was assigned 20 social contexts and 1,420 questions, randomly sampled from the benchmark. After the participants finished all the questions or decided not to proceed, we asked them to rate the complexity level and the quality of the questions and the corresponding options. In specific, we asked them to answer two questions. One is a 3-Likert scale question: (1) *complexity-1*: please rate the difficulty of the questions related to the social context, where 1 and 3 indicate easy and hard, respectively. Another

| | 📖 :Understanding | | | 🪙: Influence | | | 🔗 :Transformation | | | | | |
|---|---|---|---|---|---|---|---|---|---|---|---|---|
| Subject | Belief | | | Emotion | | | Intention | | | Action | | AVG. |
| | 📖 | 🪙 | 🔗 | 📖 | 🪙 | 🔗 | 📖 | 🪙 | 🔗 | 📖 | 🔗 | |
| Human | 0.84 | 0.85 | 0.78 | 0.90 | 0.80 | 0.79 | 0.79 | 0.72 | 0.74 | 0.77 | 0.76 | 0.78 |
| GPT-4o | 0.81 | 0.64 | 0.45 | 0.92 | 0.81 | 0.46 | 0.88 | 0.92 | 0.51 | 0.95 | 0.55 | 0.67 |
| GPT-4-Turbo | 0.63 | 0.46 | 0.33 | 0.75 | 0.53 | 0.34 | 0.72 | 0.75 | 0.35 | 0.80 | 0.37 | 0.50 |
| Llama-3.1-70B | 0.66 | 0.36 | 0.40 | 0.93 | 0.63 | 0.43 | 0.83 | 0.82 | 0.42 | 0.92 | 0.46 | 0.58 |
| Llama-3.1-8B | 0.31 | 0.28 | 0.19 | 0.39 | 0.27 | 0.19 | 0.22 | 0.25 | 0.17 | 0.27 | 0.15 | 0.23 |
| Mixtral-8x7B | 0.24 | 0.20 | 0.22 | 0.46 | 0.41 | 0.19 | 0.33 | 0.51 | 0.10 | 0.40 | 0.09 | 0.25 |
| Mistral-7B | 0.21 | 0.20 | 0.12 | 0.23 | 0.25 | 0.11 | 0.17 | 0.25 | 0.10 | 0.20 | 0.10 | 0.16 |
| Qwen2-72B | 0.72 | 0.40 | 0.38 | 0.86 | 0.65 | 0.38 | 0.80 | 0.87 | 0.34 | 0.89 | 0.20 | 0.53 |
| Qwen2-7B | 0.23 | 0.19 | 0.19 | 0.44 | 0.34 | 0.20 | 0.26 | 0.19 | 0.16 | 0.24 | 0.15 | 0.22 |
| DeepSeek-V2 | 0.07 | 0.15 | 0.10 | 0.05 | 0.10 | 0.08 | 0.04 | 0.10 | 0.07 | 0.03 | 0.06 | 0.08 |
| GLM-4 | 0.30 | 0.34 | 0.24 | 0.44 | 0.31 | 0.20 | 0.29 | 0.30 | 0.17 | 0.40 | 0.16 | 0.26 |
| LLM AVG. | 0.42 | 0.32 | 0.26 | 0.55 | 0.43 | 0.26 | 0.45 | 0.50 | 0.24 | 0.51 | 0.23 | 0.35 |
| GPT-4o+CoT | 0.79 | 0.59 | 0.44 | 0.88 | 0.72 | 0.47 | 0.82 | 0.84 | 0.47 | 0.90 | 0.50 | 0.63 |
| GPT-4-Turbo+CoT | 0.61 | 0.46 | 0.30 | 0.78 | 0.52 | 0.33 | 0.72 | 0.69 | 0.32 | 0.81 | 0.37 | 0.49 |
| Llama-3.1-70B+CoT | 0.68 | 0.40 | 0.39 | 0.91 | 0.62 | 0.43 | 0.82 | 0.79 | 0.42 | 0.96 | 0.46 | 0.58 |
| Llama-3.1-8B+CoT | 0.31 | 0.28 | 0.21 | 0.40 | 0.25 | 0.21 | 0.21 | 0.25 | 0.20 | 0.24 | 0.16 | 0.24 |
| Mixtral-8x7B+CoT | 0.16 | 0.16 | 0.14 | 0.29 | 0.27 | 0.13 | 0.25 | 0.32 | 0.09 | 0.26 | 0.08 | 0.18 |
| Mistral-7B+CoT | 0.21 | 0.21 | 0.11 | 0.22 | 0.25 | 0.11 | 0.20 | 0.25 | 0.09 | 0.19 | 0.09 | 0.16 |
| Qwen2-72B+CoT | 0.71 | 0.38 | 0.40 | 0.87 | 0.68 | 0.41 | 0.83 | 0.87 | 0.35 | 0.88 | 0.27 | 0.55 |
| Qwen2-7B+CoT | 0.28 | 0.17 | 0.18 | 0.43 | 0.36 | 0.19 | 0.30 | 0.22 | 0.19 | 0.20 | 0.18 | 0.23 |
| DeepSeek-V2+CoT | 0.08 | 0.17 | 0.09 | 0.04 | 0.11 | 0.10 | 0.05 | 0.13 | 0.07 | 0.05 | 0.06 | 0.09 |
| GLM-4+CoT | 0.30 | 0.36 | 0.26 | 0.48 | 0.30 | 0.22 | 0.33 | 0.31 | 0.17 | 0.43 | 0.15 | 0.28 |
| LLM+CoT AVG. | 0.41 | 0.32 | 0.25 | 0.53 | 0.41 | 0.26 | 0.45 | 0.47 | 0.24 | 0.49 | 0.23 | 0.34 |

Table 3: LLMs' performances. We show the performance according to the combination of ToM reasoning items and question types. The items include belief, emotion, intention, and action. The question types include understanding, influence, and transformation. "LLM AVG." and "LLM+CoT AVG." are the average performance of all the ten LLMs and LLMs+CoT, respectively. The best performance of vanilla prompting is in purple, and that of CoT prompting is in grey.

is a 5-Likert scale question: (2) *complexity-2*: please rate the quality of the questions and options. 1 is "(Bad) all questions do not make sense," while 5 is "(Perfect) Solvable, not too trivial."

Figure 3 presents the results. We observe that the questions have varying complexity levels: 23.8% of the questions can be quickly solved by most annotators, 69% require some effort, and 7.2% can only be solved by professionals or with great effort.

More details about the human evaluation of social contexts and questions in the benchmark can be found in Appendix A.6.

# 5 EXPERIMENTS

## 5.1 EXPERIMENTAL SETUP

We use TOMVALLEY to evaluate the ToM abilities of 10 popular LLMs, including GPT-4o (OpenAI, 2024), GPT-4-Turbo (Achiam et al., 2023), Llama-3.1-8B (AI@Meta, 2024), Llama-3.1-70B (AI@Meta, 2024), Mistral-7B (AI, 2024a), Mixtral-8x7B (AI, 2024b), Qwen2-7B (Yang et al., 2024), Qwen2-72B (Yang et al., 2024), DeepSeek-V2 (Liu et al., 2024a), GLM-4 (GLM et al., 2024). We strictly abide by all LLMs' terms and get access through official APIs or model weights. We employ two prompting methods: vanilla prompting directly asking LLMs to answer the questions, and CoT prompting elicits step-by-step reasoning before answering. A human baseline is established by inviting five graduate students to complete a subset of TOMVALLEY. Details about model versions, parameter sizes, context window sizes, and the prompts used for the two methods are shown in Appendix B.

| | ToM Reasoning Item Type | | | | Question Type | | |
|---|---|---|---|---|---|---|---|
| | Belief | Emotion | Intention | Action | Understanding | Influence | Transformation |
| Human | 0.81 | 0.81 | 0.73 | 0.75 | 0.82 | 0.79 | 0.77 |
| GPT-4o | 0.59 | 0.68 | 0.72 | 0.69 | 0.89 | 0.79 | 0.49 |
| GPT-4-Turbo | 0.44 | 0.50 | 0.55 | 0.52 | 0.73 | 0.58 | 0.35 |
| Llama-3.1-70B | 0.46 | 0.61 | 0.63 | 0.63 | 0.84 | 0.60 | 0.43 |
| Llama-3.1-8B | 0.24 | 0.26 | 0.21 | 0.19 | 0.30 | 0.27 | 0.17 |
| Mixtral-8x7B | 0.22 | 0.32 | 0.27 | 0.20 | 0.36 | 0.37 | 0.15 |
| Mistral-7B | 0.16 | 0.18 | 0.16 | 0.14 | 0.20 | 0.23 | 0.11 |
| Qwen2-72B | 0.47 | 0.58 | 0.60 | 0.45 | 0.81 | 0.64 | 0.33 |
| Qwen2-7B | 0.20 | 0.30 | 0.20 | 0.19 | 0.29 | 0.24 | 0.18 |
| GLM-4 | 0.28 | 0.29 | 0.23 | 0.24 | 0.36 | 0.31 | 0.19 |
| LLM AVG. | 0.34 | 0.41 | 0.40 | 0.36 | 0.53 | 0.45 | 0.27 |

Table 4: LLMs' performance in vanilla prompting. We show the performance according to ToM reasoning items and question types, respectively. Due to space limitations, we don't show the results in CoT prompting, which is shown in Appendix B.3.

## 5.2 MAIN RESULTS

Table 3 and 4 demonstrate the ToM performances of LLMs according to ToM reasoning items and question types. As mentioned, the ToM reasoning items include belief, emotion, intention, and action; and the question types include understanding, influence, and transformation. We discuss the results and highlight several critical observations as follows.

**Human vs. LLMs**  Humans achieve 78% accuracy performance. However, ToM performances of all LLMs are significantly lower, with the smallest gap being 11% in vanilla prompting (Human 78% vs. GPT-4o 67%). Among the question types, LLMs fall behind humans in transformation in all ToM reasoning items, representing the most challenging reasoning for LLMs. Interestingly, in the understanding type, LLMs like GPT-4o even outperform humans, which we believe is explainable. As shown in Figure 1, the understanding question mainly directly asks what the mental state of the character is in one scenario, which requires less reasoning process compared to other questions, and LLMs can easily answer these questions by semantic matching.

**LLMs' ToM Performance**  In the vanilla prompting, GPT-4o stands out in LLMs and surpasses the second of Llama-3.1-70B up to 9% points. Among the open LLMs, Llama-3.1-70B and Qwen2-72B impressively outperforms other LLMs and even outperforms GPT-4-Turbo. The Llama-3.1-70B also surpasses the GPT-4o in the understanding type question of emotion ability. However, even the most superior GPT-4o only reaches 67%, and the lowest score is 8%, which shows the difficulty of our benchmark, and current LLMs lack robust ToM reasoning in the social context environment, even if they reach nearly 100% performance in other benchmarks (Gandhi et al., 2024).

**Vanilla vs. CoT Prompting**  Both ability and question-type results indicate that CoT prompting doesn't always improve LLMs' ToM reasoning ability. The CoT successfully improved their performance for Llama-3.1-8B, Qwen2-series, DeepSeek-V2, and GLM-4. For other models, such as GPT-4o, the CoT prompting even leads to a decline in performance. This finding aligns with the findings in Xiao et al. (2023); Chen et al. (2024). One plausible explanation is that CoT reasoning predominantly works by deconstructing intricate problems into more easy sub-tasks. Nonetheless, CoT cannot help to improve basic ToM ability. So, CoT will only work for models that do not have the ability to decompose complex problems. We present a failure case of GPT-4o when CoT prompting is used in Appendix B.5.

**Differences Across ToM Reasoning Items**  Exploring Table 4, the best-performing ToM reasoning item is emotion, consistent with the results found in TOMBENCH (Chen et al., 2024). Belief is the weakest item, trailing emotion by 7%. This demonstrates that the model is less proficient in addressing belief-related issues than the other 3 items. One potential explanation is that the character's belief is more concealed than other mental states, necessitating the model to perform additional

reasoning steps, particularly when the questions necessitate the model to infer the character's belief from observed actions, which align with the findings in BigToM (Gandhi et al., 2024).

**Differences Across Question Types** Further exploring Table 4, the model performs poorly in influence and transformation types compared with understanding, especially the question type of transformation, which lags behind understanding by 26 percent. The transformation question mainly relates to mental states' transformation in different scenarios, such as how the character's mental state changes from one scenario to another. This shows that the models lack the ability to handle the mental states in the middle scenarios of social context, where the dynamism of the mental state across scenarios is inherent in the social interaction.

**Profile Absence vs. Profile Presence** As stated by previous studies, personal profile information and the social norms behind the social location can help human to accurately ascertain an individual's mental state (Bretherton & Beeghly, 1982; Strang, 1930). To further examine whether LLMs' ToM performance will be influenced by the absence of this kind of information, we randomly pick 100 social contexts, a total of 7100 questions, and evaluate GPT-4o's performance with and without the characters' profile as part of the input, denoted as presence and absence respectively. As shown in Figure 4, the model lags behind 7 percent at most when there is no profile as part of the input, which aligns with the findings in the psychological literature. Most current work does not include such information, so their benchmarks can only evaluate LLMs' performance

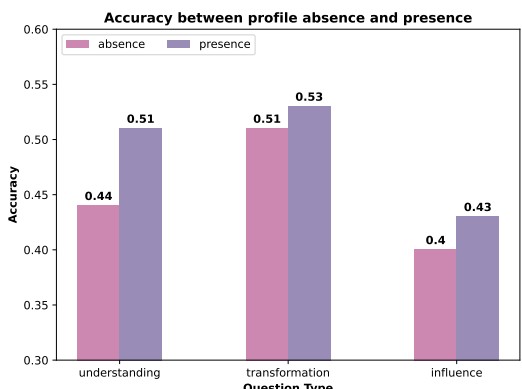

Figure 4: The accuracy of GPT-4o in the 3 question types with the presence of profile and absence of profile.

without such information. Our benchmark allowing us to more completely evaluate the model's performance in various conditions.

## 5.3 IN-DEPTH ANALYSIS

**LLMs Fail in the Middle Scenario** In Table 4, we find that the model performs poorly in transformation type compared with understanding and influence for all the models. To further explore the reason why models perform poorly when handling the transformation of ToM reasoning items, we further show the models' performance of transformation along the time span. The time span indicates the specific scenarios to which one question relates. For example, the type 3 question in Figure 1 relates scenarios 1 and 2 and the ToM reasoning item of belief. As shown in Figure 5, the model performs better in the early and the end scenarios, while the model performs worse in the middle scenario. We posit that this may result from the model's "Lost in the middle" phenomenon (Liu et al., 2024b): models exhibit diminished performance when crucial informa-

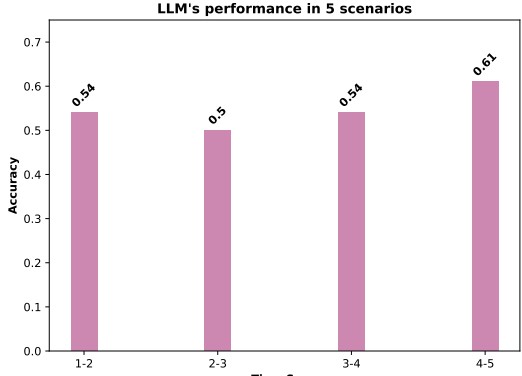

Figure 5: The average of models' scores in the transformation question type. The time span indicates the specific scenarios to which one question relates.

tion is retrieved from extensive contexts, with optimal performance typically occurring at the beginning or end of the input context. The social scenario is presented to LLMs from scenarios 1 to 5; hence, the time order correlates with the position in the input, and LLMs perform poorly for the

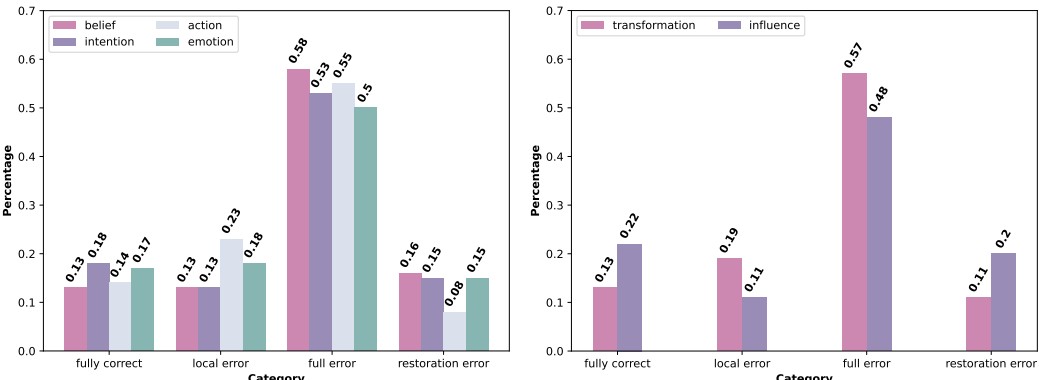

Figure 6: The percentage of four status of the compositional problem. The left one is depicted in accordance with ToM reasoning items. The right one is demonstrated through the question types.

middle time span. To confirm this conjecture, we further construct 15 social contexts with 4, 6, and 7 scenarios, respectively. The results are shown in Appendix B.4, which confirms our speculation.

**LLMs' Limits of ToM on Compositionality**  Our benchmark includes a kind of compositional problem requiring rigorous multi-hop reasoning to reach the answer (Dziri et al., 2024). Utilize the type 3 illustrated in Figure 1 as a reference. To address this question, it is essential to first ascertain the character's beliefs in scenarios 1 and 2 (referred to as dependency D), followed by identifying the potential factors that induce the transformation of belief (denoted as C). Usually, there are four statuses (Dziri et al., 2024) of C: (1) **fully correct**: LLMs correctly answer both D and C. (2) **local error** LLMs only correctly answer D and wrongly answer C. (3) **restoration error**: LLMs correctly answer C but wrongly answer D. (4) **full error**: LLMs wrongly answer both D and C. Examining the failure modes of LLMs in this kind of question might elucidate if models genuinely develop ToM reasoning capability or merely depend on shortcut learning through pattern matching to answer questions. In Figure 6, we visualize the proportion of the four statuses of GPT-4o. Notably, the figure does not display the understanding questions, as they typically function as dependence problems. Of the four statuses of C, the restoration error constitutes approximately 15%, indicating that LLMs may respond to certain questions based on superficial patterns rather than via logical reasoning. The fully correct status only accounts for approximately 15%, indicating that our benchmark's compositional questions pose significant challenges for LLMs. The fully correct status of the transformation questions accounts for merely 13%, which is lower than that of influence questions; this implies that LLMs are less proficient at deducing the alterations of the ToM reasoning items.

# 6 CONCLUSION

In this paper, we propose TOMVALLEY, a benchmark to evaluate LLMs' ToM ability in a realistic social context. Different from most previous benchmarks for ToM, TOMVALLEY evaluates LLMs with three characteristics: (1) Dynamic mental states. Rather than merely assessing the final mental state after several social scenarios, TOMVALLEY incorporates questions exploring mental state alterations across multiple continuous social scenarios. (2) Intradependent mental states. TOM-VALLEY investigates how different types of mental states influence one another. (3) Provision of social location and character profiles. Feedback from annotators indicates that our evaluation data closely resembles real-life interactions. Experimental results show that current LLMs underperform humans significantly on the ToM capability: even the best LLM's performance is lower than humans' by 11%. Further analysis shows that LLMs' performance will decrease when useful information is presented in the middle of input. Moreover, we find that LLMs are incapable of processing compositional problems. With the development of LLMs, they have played an important role in constructing AI applications, such as embodied intelligence and AI agents. We hope that TOMVALLEY will drive the evaluation of ToM in closer alignment with the requirements in real social interactions, facilitating the improvement of LLMs with better ToM abilities.

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

# A THE CONSTRUCTION OF THE TOMVALLEY

## A.1 THE CANDIDATE POOL OF SOCIAL LOCATION

The social location describes the environments where individuals live, work, and learn, which can significantly impact their mental states and behavior (Stokols, 1978). As shown in Figure 7, we have collected 13 types of social location types in total, adding up to 261 locations, which is referred from Ziems et al. (2023).

## A.2 THE CANDIDATE POOL OF PROFILE

We conclude 7 aspects in the characters' profile: surname, name, gender, occupation, education, race, and personality traits. Their value can be found in Figure 9, 10, 11, and 12. The source of the surname, name, and occupation statistics are U.S. Census Bureau Homepage, The United States Social Security Administration, and Bureau of Labor Statistics, respectively. Figure 8 shows an example of the social background.

## A.3 THE PROMPT USED TO GENERATE THE SKETCH OF MENTAL STATES

As illustrated in Figure 13, the prompt is used to generate the sketch of mental states. In the holders of '{}' and '[]', the corresponding information will be input into this prompt. An example of the sketch of mental states is shown in Figure 14.

| Model | Version | Size | Context Length |
|---|---|---|---|
| GPT-4o | 2024-05-13 | ~ | 128k |
| GPT-4-Turbo | 2024-04-09 | ~ | 128k |
| Llama-3.1-8B | Instruct | 8B | 128k |
| Llama-3.1-70B | Instruct | 70B | 128k |
| Mistral-7B | Instruct-v0.3 | 7B | 32k |
| Mixtral-8x7B | Instruct-v0.1 | 8x7B | 32k |
| Qwen2-7B | Instruct | 7B | 128k |
| Qwen2-72B | Instruct | 72B | 128k |
| DeepSeek-V2 | Lite-Chat | 16B | 32k |
| GLM-4 | 9b-chat | 9B | 128k |

Table 5: The detail of models evaluated in our benchmark.

### A.4 THE PROMPT USED TO GENERATE THE SOCIAL SCENARIOS

As illustrated in Figure 15, the prompt is used to generate the social scenarios. In the holders of '{}' and '[]', the corresponding information will be input into this prompt. An example of the social scenario is shown in Figure 16.

### A.5 THE TEMPLATES FOR THE FIVE TYPES OF QUESTIONS AND QUESTION EXAMPLE

we will apply 5 predefined question templates to the social context to generate 5 types of questions, 71 questions for every social context in total. The five types of questions are: (1) (Understanding-1) What is the main character's ToM reasoning item in a specific scenario? (2) (Influence-1) In one scenario, how does mental state A influence ToM reasoning item B? (3) (Transformation-1) Does a ToM reasoning item change from scenario A to scenario B? (4) (Transformation-2) What causes a ToM reasoning item change from scenario A to scenario B? (5) (Transformation-3) How does the ToM reasoning item change across all the scenarios? The templates and the example of the five types of questions are shown in Figure 17.

### A.6 HUMAN EVALUATION OF THE QUALITY OF TOMVALLEY

We apply argilla as the annotation platform. The annotation interface for the social context quality evaluation is shown in Figure 18. The annotation interface for the question quality and qomplexity evaluation is shown in Figure 19. We invite five graduate student volunteers for the human evaluation and the subsequent human baseline. Volunteers are paid $10.48/hr (amount converted by exchange rate).

## B EXPERIMENTS

### B.1 MODEL DETAIL

We evaluate a total of 10 popular LLMs, including GPT-4o (OpenAI, 2024), GPT-4-Turbo (Achiam et al., 2023), Llama-3.1-8B (AI@Meta, 2024), Llama-3.1-70B (AI@Meta, 2024), Mistral-7B (AI, 2024a), Mixtral-8x7B (AI, 2024b), Qwen2-7B (Yang et al., 2024), Qwen2-72B (Yang et al., 2024), DeepSeek-V2 (Liu et al., 2024a), GLM-4 (GLM et al., 2024). For all the LLMs, we strictly abide by their terms and get access through official APIs or model weights. Details about model versions, parameter sizes, context window sizes and the prompts used for the two methods are shown in Table 5.

### B.2 PROMPTING METHODS

We employ two prompting methods: the vanilla prompting which directly asks LLMs to answer the questions, and the CoT prompting that elicits step-by-step reasoning before answering. The prompts used for the two methods are shown in Figure 20.

### B.3 LLMs' PERFORMANCE IN COT PROMPTING

Here, we show the performance of LLMs in CoT prompting according to ToM reasoning items and question types, respectively. The results are shown in Table 6.

### B.4 TRANSFORMATION PERFORMANCE

In Section 5.3, we find that the model performs better in the early and the end scenarios, while the model performs worse in the middle scenario. To confirm that this may result from the model's "Lost in the middle" phenomenon, we further construct 15 social contexts with 4, 6, and 7 scenarios, respectively. The results are shown in Figure 21. Furthermore, we ran a comparative experiment to negate the potential that questions about the middle scenario are more challenging than others, resulting in inferior model performance. For the social context with 5, 6, and 7 scenarios, we only keep the first four scenarios and compare the performance of GPT-4o in the first three time spans (1-2,2-3 and 3-4) with its performance when the remaining scenarios are not truncated. As shown in Figure 22, almost all the performance in middle spans has been improved when the last scenario(s) are truncated. This further confirms our speculation.

### B.5 CASE STUDY FOR COT PROMPTING

Both ToM reasoning item and question-type results in Table 3 indicate that CoT prompting doesn't always improve LLMs' ToM reasoning ability. We present a failure case of GPT-4o when CoT prompting is used in Figure 23.

## C  LIMITATIONS, FUTURE DIRECTIONS, ANDFUTURE DIRECTIONS

We discuss the limitations, ethical considerations, and future directions below.

**Limitations and future directions.**    In ToMValley, we aim to evaluate LLMs' ToM in a realistic social context. To make up the social context, we collect the candidate pool of social locations and profiles. we collect 261 locations. For the aspect of surname, name, and occupations for profile, there are 100 candidates for every aspect. Even though the combination of these would outcome diverse social contexts, it is still limited compared with the diversity level of social contexts in real life. Besides, perception is an important mental state that helps humans capture social signals from outer environments. Because we mainly focus on evaluating LLMs that cannot process image or video input, we do not include perception as the mental state in our benchmark. Additionally, we evaluate 10 popular LLMs in the experiments. Due to the cost of running inference of commercial LLMs, we only evaluate the GPT-series. The evaluation for other commercial LLMs, such as Claude, could be included in future analyses. We use vanilla and CoT prompting methods for evaluation, while other prompting methods, such as "Tree of thoughts" (Yao et al., 2024), could also be explored to determine their effect on ToM abilities. Moreover, large language models trained on online content unavoidably acquire stereotypical associations related to gender, ethnicity, and other characteristics. This may result in normative, stereotypical effects of LLMs for generations (Gandhi et al., 2024). So, when evaluating the LLMs' ToM ability, their performance can be influenced by the identity information in the profile. Future work could further investigate whether the identity information in the profile can affect the LLMs' ToM performance. For example, when we only alter the race or surname in the profile, leaving other information remaining, whether the LLMs' ToM performance will change accordingly or not. Identifying potential biases of LLMs' performance towards different profile information in ToMValley could also help researchers to better improve LLMs' ToM reasoning ability.

**Ethical Considerations.**    The theory of mind is a distinctive social cognitive capability that is intrinsic to humans. Assessing the Theory of Mind capacities of Large Language Models utilizing ToMValley may result in anthropomorphic interpretations, attributing human-like qualities to LLMs. Nonetheless, it is imperative to clarify that our objective is not to anthropomorphize LLMs. Our objective is to evaluate the capacity of LLMs to comprehend and interpret human mental states, thus enhancing AI's interaction with humans in the social context.

**The Candidate Pool of Social Location**

**Transportation and Travel**
airplane cabin, airplane, airport terminal, airport, bus stop, bus boat, car, helicopter, harbor, dock, hot air balloon, ship, taxi, train compartment, train station, subway platform, subway

**Accommodation and Residential**
apartment, backyard, balcony, beach house, bedroom, cabin campsite, childs room, dorm room, hotel lobby, hotel pool, hotel room hotel, house entrance, house safe, living room, nursery, nursing home, private deck, residential neighborhood, yard

**Food and Beverage**
bakery, bar, cafe, cafeteria, deli restaurant, dining hall, fastfood restaurant, outdoor restaurant, pub, restaurant, steak house

**Shopping and Retail**
arcade, bookstore, boutique, camera shop, car dealership, clothing store, department store, drugstore, gift shop, grocery store, hardware store, jewelry shop, liquor store, market, music store, pawn shop, shop, shopping mall, showroom, store

**Entertainment and Leisure**
amusement park, arcade, balloon festival, ballroom, cinema, circus, fair, game room, golf course, museum, nightclub, outdoor cinema, paintball park, park, performance hall, ski resort, theater, yoga studio, zoo

**Education**
classroom, college, kindergarten, biology laboratory, lab, library, school bus, school theater, school, university

**Health and Wellness**
alcoholics ward, clinic, dentists office, doctors office, hospital, mental institution, nurse room, veterinarians office

**Work and Office**
bank, campaign room, conference room, construction site, copy center, government office, home office, office

**Military and Law Enforcement**
border, police car, police station, border checkpoint

**Places of Worship and Ceremony**
altar, chapel, church, funeral parlour, temple

**Nature and Outdoors**
canyon, cliff, coast, cornfield, desert, forest, lake, lawn, meadow, mountain, ocean, plain, river, waterfall, woods

**Sport and Fitness**
athletic field, basketball court, gym, gymnasium, hockey rink, locker room, mini golf course, polo field, pool, riding arena, running track, soccer field, stadium, tennis court, volleyball field

**Miscellaneous**
animal shelter, arena, art gallery, attic, auditorium, backyard, basement, bathroom, bazaar, beach, beauty salon, boathouse, bowling alley, boxing hall, bridal shop, bridge, broadcast studio, castle, cave, cemetery, city hall, city plaza, cloakroom, closet, clubhouse, community center, country road, courthouse, courtyard, dance studio, darkroom, drawing room, elevator, excavation, exhibition hall, farm, film studio, football field, garage, garden, gas station, gun shop, hall, hangar, hawaii, hotel, junkyard, kitchen table, kitchen, lake, legislative chamber, liquor store, living room, market, massage parlor, monument, museum, newsroom, outdoor pool, patio, performance hall, plaza, political rally, porch, post office, repair shop, restroom, road, rodeo, rooftop, rural road, saloon, smoking room, sofa, stable, stage, study room, tattoo studio, telephone booth, tv studio, waiting room

Figure 7: The candidate pool of social location.

**An example of the social background**

**social setting**: "temple"

**social setting type**: "Places of Worship and Ceremony"

**main character**: "Maria Thai"

**scenario numbers**: 5

**characters information**: "**Main Character**: Maria Thai, a woman, is a Social Worker whose race is Native Hawaiian and Other Pacific Islander. Maria Thai obtained a doctorate degree and has an ESFP personality..**Supporting Characters**:- **1**: Kelly Sexton, the woman, is a Fitness Trainer whose race is White. Kelly Sexton obtained a Primary Education and has an ISTP personality."

Figure 8: An example of the social background.

**The Candidate Pool of Surname**

**American Indian and Alaska Native**
Begay, Yazzie, Benally, Tsosie, Nez, Begaye, Etsitty, Becenti, Yellowhair, Manygoats, Wauneka, Manuelito, Apachito, Bedonie, Calabaza, Peshlakai, Claw, Roanhorse, Goldtooth, Etcitty, Tsinnijinnie, Notah, Clah, Atcitty, Twobulls, Werito, Hosteen, Yellowman, Attakai, Bitsui, Delgarito, Henio, Goseyun, Keams, Secatero, Declay, Tapaha, Beyale, Haskie, Cayaditto, Blackhorse, Ethelbah, Tsinnie, Walkingeagle, Altaha, Bitsilly, Wassillie, Benallie, Smallcanyon, Littledog, Cosay, Clitso, Tessay, Secody, Bigcrow, Tabaha, Chasinghawk, Blueeyes, Olanna, Blackgoat, Cowboy, Kanuho, Shije, Gishie, Littlelight, Laughing, Whitehat, Eriacho, Runningcrane, Chinana, Kameroff, Spottedhorse, Arcoren, Whiteplume, Dayzie, Spottedeagle, Heavyrunner, Standingrock, Poorbear, Ganadonegro, Ayze, Whiteface, Yepa, Talayumptewa, Madplume, Bitsuie, Tsethlikai, Ahasteen, Dosela, Birdinground, Todacheenie, Bitsie, Todacheene, Bullbear, Lasiloo, Keyonnie, Notafraid, Colelay, Kallestewa, Littlewhiteman

**Asian and Native Hawaiian and Other Pacific Islander**
Nguyen, Kim, Patel, Tran, Chen, Li, Le, Wang, Yang, Pham, Lin, Liu, Huang, Wu, Zhang, Shah, Huynh, Yu, Choi, Ho, Kaur, Vang, Chung, Truong, Phan, Xiong, Lim, Vo, Vu, Lu, Tang, Cho, Ngo, Cheng, Kang, Tan, Ng, Dang, Do, Ly, Han, Hoang, Bui, Sharma, Chu, Ma, Xu, Zheng, Song, Duong, Liang, Sun, Zhou, Thao, Zhao, Shin, Zhu, Leung, Hu, Jiang, Lai, Gupta, Cheung, Desai, Oh, Ha, Cao, Yi, Hwang, Lo, Dinh, Hsu, Chau, Yoon, Luu, Trinh, He, Her, Luong, Mehta, Moua, Tam, Ko, Kwon, Yoo, Chiu, Su, Shen, Pan, Dong, Begum, Gao, Guo, Chowdhury, Vue, Thai, Jain, Lor, Yan, Dao

**Black or African American**
Smalls, Jeanbaptiste, Diallo, Kamara, Pierrelouis, Gadson, Jeanlouis, Bah, Desir, Mensah, Boykins, Chery, Jeanpierre, Boateng, Owusu, Jama, Jalloh, Sesay, Ndiaye, Abdullahi, Wigfall, Bienaime, Diop, Edouard, Toure, Grandberry, Fluellen, Manigault, Abebe, Sow, Traore, Mondesir, Okafor, Bangura, Louissaint, Cisse, Osei, Calixte, Cephas, Belizaire, Fofana, Koroma, Conteh, Straughter, Jeancharles, Mwangi, Kebede, Mohamud, Prioleau, Yeboah, Appiah, Ajayi, Asante, Filsaime, Hardnett, Hyppolite, Saintlouis, Jeanfrancois, Ravenell, Keita, Bekele, Tadesse, Mayweather, Okeke, Asare, Ulysse, Saintil, Tesfaye, Jeanjacques, Ojo, Nwosu, Okoro, Fobbs, Kidane, Petitfrere, Yohannes, Warsame, Lawal, Desta, Veasley, Addo, Leaks, Gueye, Mekonnen, Stfleur, Balogun, Adjei, Opoku, Coaxum, Vassell, Prophete, Lesane, Metellus, Exantus, Hailu, Dorvil, Frimpong, Berhane, Njoroge, Beyene

**Hispanic or Latino**
Garcia, Rodriguez, Martinez, Hernandez, Lopez, Gonzalez, Perez, Sanchez, Ramirez, Torres, Flores, Rivera, Gomez, Diaz, Morales, Gutierrez, Ortiz, Chavez, Ruiz, Alvarez, Castillo, Jimenez, Vasquez, Moreno, Herrera, Medina, Aguilar, Vargas, Guzman, Mendez, Munoz, Salazar, Garza, Soto, Vazquez, Alvarado, Delgado, Pena, Contreras, Sandoval, Guerrero, Rios, Estrada, Ortega, Nunez, Maldonado, Dominguez, Vega, Espinoza, Rojas, Marquez, Padilla, Mejia, Juarez, Figueroa, Avila, Molina, Campos, Ayala, Carrillo, Cabrera, Lara, Robles, Cervantes, Solis, Salinas, Fuentes, Velasquez, Aguirre, Ochoa, Cardenas, Calderon, Rivas, Serrano, Rosales, Castaneda, Gallegos, Ibarra, Suarez, Orozco, Salas, Escobar, Velazquez, Macias, Zamora, Villarreal, Barrera, Pineda, Santana, Trevino, Lozano, Rangel, Arias, Mora, Valenzuela, Zuniga, Melendez, Galvan, Velez, Meza

**White**
Olson, Snyder, Wagner, Meyer, Schmidt, Ryan, Hansen, Hoffman, Johnston, Larson, Carlson, Obrien, Jensen, Hanson, Weber, Walsh, Schultz, Schneider, Keller, Beck, Schwartz, Becker, Wolfe, Zimmerman, Mccarthy, Erickson, Klein, Oconnor, Swanson, Christensen, Fischer, Wolf, Gallagher, Schroeder, Parsons, Bauer, Mueller, Hartman, Kramer, Flynn, Owen, Shaffer, Hess, Olsen, Petersen, Roth, Hoover, Weiss, Decker, Yoder, Larsen, Sweeney, Foley, Hensley, Huffman, Cline, Oneill, Koch, Brennan, Berg, Russo, Macdonald, Kline, Jacobson, Berger, Blankenship, Bartlett, Odonnell, Stein, Stout, Sexton, Nielsen, Howe, Morse, Knapp, Herman, Stark, Hebert, Schaefer, Reilly, Conrad, Donovan, Mahoney, Hahn, Peck, Boyle, Hurley, Mayer, Mcmahon, Case, Duffy, Friedman, Fry, Dougherty, Crane, Huber, Moyer, Krueger, Rasmussen, Brandt

Figure 9: The races and their corresponding 100 most popular surnames.

**The Candidate Pool of name**

**Man**
James, Michael, Robert, John, David, William, Richard, Joseph, Thomas, Christopher, Charles, Daniel, Matthew, Anthony, Mark, Donald, Steven, Andrew, Paul, Joshua, Kenneth, Kevin, Brian, Timothy, Ronald, George, Jason, Edward, Jeffrey, Ryan, Jacob, Nicholas, Gary, Eric, Jonathan, Stephen, Larry, Justin, Scott, Brandon, Benjamin, Samuel, Gregory, Alexander, Patrick, Frank, Raymond, Jack, Dennis, Jerry, Tyler, Aaron, Jose, Adam, Nathan, Henry, Zachary, Douglas, Peter, Kyle, Noah, Ethan, Jeremy, Christian, Walter, Keith, Austin, Roger, Terry, Sean, Gerald, Carl, Dylan, Harold, Jordan, Jesse, Bryan, Lawrence, Arthur, Gabriel, Bruce, Logan, Billy, Joe, Alan, Juan, Elijah, Willie, Albert, Wayne, Randy, Mason, Vincent, Liam, Roy, Bobby, Caleb, Bradley, Russell, Lucas

**Woman**
Mary, Patricia, Jennifer, Linda, Elizabeth, Barbara, Susan, Jessica, Karen, Sarah, Lisa, Nancy, Sandra, Betty, Ashley, Emily, Kimberly, Margaret, Donna, Michelle, Carol, Amanda, Melissa, Deborah, Stephanie, Rebecca, Sharon, Laura, Cynthia, Dorothy, Amy, Kathleen, Angela, Shirley, Emma, Brenda, Pamela, Nicole, Anna, Samantha, Katherine, Christine, Debra, Rachel, Carolyn, Janet, Maria, Olivia, Heather, Helen, Catherine, Diane, Julie, Victoria, Joyce, Lauren, Kelly, Christina, Ruth, Joan, Virginia, Judith, Evelyn, Hannah, Andrea, Megan, Cheryl, Jacqueline, Madison, Teresa, Abigail, Sophia, Martha, Sara, Gloria, Janice, Kathryn, Ann, Isabella, Judy, Charlotte, Julia, Grace, Amber, Alice, Jean, Denise, Frances, Danielle, Marilyn, Natalie, Beverly, Diana, Brittany, Theresa, Kayla, Alexis, Doris, Lori, Tiffany

Figure 10: The genders and their corresponding 100 most popular names.

**The Candidate Pool of occupation**

**Man**

Construction Worker, Electrician, Plumber, Carpenter, Mechanic, Software Developer, Engineer, Pilot, Truck Driver, Police Officer, Firefighter, Doctor, Lawyer, Architect, Welder, Heavy Equipment Operator, Factory Worker, Farmer, Security Guard, Salesperson, Financial Analyst, Accountant, Real Estate Agent, Chef, IT Support Specialist, Project Manager, Business Analyst, Marketing Manager, Entrepreneur, Military Personnel, Maintenance Worker, Landscaper, Sports Coach, Athlete, Bartender, Warehouse Worker, Janitor, Civil Engineer, Graphic Designer, HVAC Technician, Delivery Driver, Scientist, Researcher, College Professor, Banker, Insurance Agent, Surveyor, Veterinarian, Barber, Personal Trainer

**Woman**

Teacher, Nurse, Receptionist, Administrative Assistant, Customer Service Representative, Cashier, Retail, Salesperson, Waitress, Medical Assistant, Hairdresser, Childcare Worker, Social Worker, Librarian, Human Resources Manager, Marketing Manager, Accountant, Dental Hygienist, Pharmacist, Flight Attendant, Secretary, Occupational Therapist, Physical Therapist, Registered Nurse, Home Health Aide, Event Planner, Public Relations Specialist, Dietitian, Psychologist, Interior Designer, Speech-Language Pathologist, Elementary School Teacher, Fashion, Designer, Real Estate Agent, Paralegal, Personal Care Aide, Health Services Manager, Legal Secretary, Bookkeeper, Preschool Teacher, Technical Writer, Graphic Designer, Midwife, Bartender, Bank Teller, Office Manager, Translator, Fitness Trainer, Editor, Market Research Analyst, Sales Representative

Figure 11: The genders and their corresponding 100 most popular occupations.

**The Candidate Pool of Personality Trait and Education**

**Personality Traits**

istj: "The Inspector", isfj: "The Protector", infj: "The Counselor", intj: "The Mastermind"
istp: "The Craftsman", isfp: "The Composer", infp: "The Healer", intp: "The Architect"
estp: "The Dynamo", esfp: "The Performer", enfp: "The Champion", entp: "The Visionary"
estj: "The Supervisor", esfj: "The Provider", enfj: "The Teacher", entj: "The Commander"

**Educations**

primary education, middle school, high school, bachelor's degree, master's degree, doctorate degree

Figure 12: The personality traits and educations.

**Prompt for Sketch of Mental States**

"[character information]:{characters_information}\n I want to generate a complete and complex story that happened in {social_setting}. The story includes {scenario_number} scenarios. Every scenario, including conversation and the background of the conversation between {main_character} and supporting character. These scenarios are to test the readers' theory of mind reasoning abilities in four mental states: belief, emotion, intention, and action. The four mental states will influence each other: a, belief will influence emotion; b, belief and emotion will influence intention; c, intention will influence action. You do not need to generate the complete story and scenarios. But you could help me to: 1. Describe the relationship between the {main_character} and the other provided character.2.design what the belief, emotion, intention, and action are of the {main_character} in every scenario when {main_character} interacts with supporting characters. 3. The previous scenario's mental states may influence the following scenarios' mental states. The changes in mental states are caused both by previous mental states and the plot. 4. I want the mental states' change to be complex enough: at least one mental state of [belief, emotion, intention] changes in all scenarios. You should also describe how each mental state influences the other.The json format should be:{{[scenario number]:{{[belief]:[content],[emotion]:[content],[intention]:[content],[action]:[content],[influence]:{{[belief->emotion]:[content],[belief&emotion->intention]:[content],[intention->action]:[content] }} }} }} 5. After steps 1-4, you should analyze all mental states of {main_character}: if changed between scenarios, you should also provide the main plot and previous mental states that cause the mental states' change. The format should be: [belief]: [if changed: True or False], [if changed] [scenario number]:[mental state's value]->[scenario number]:[mental state's value]->[scenario number]:[mental state's value] …Reasons: [scenario number]->[scenario number]:[reason];[scenario number]->[scenario number]:[reason]…[emotion]: the format is the same as [belief].[intention]: the format is the same as [belief].[action]: the format is the same as [belief] 6. the interaction in every scenario should be between the {main_character} and supporting characters. 7. Respond with JSON format:{JSON_format}. 8. these scenarios are very hard and complex, composed of severe plot conflicts and mental state changes, so that even adults will have to take a long time or seek help from others to understand these scenarios."

Figure 13: The prompt for the generation of the relationship between characters and the sketch of mental states.

```
                          Example of Mental Sketch

                         relationships among characters

"John Atcitty and Beverly Yan are colleagues at a community center where John teaches equipment operation and
Beverly teaches adult education classes. They share a mutual respect for each other's skills and often collaborate
on projects that require both practical and theoretical knowledge."

                       mental states analysis in every scenario

scenario 1:
    "belief": "John believes Beverly might think less of him due to his lower level of formal education."
    "emotion": "John feels insecure and slightly anxious."
    "intention": "John intends to prove his competence."
    "action": "John goes out of his way to demonstrate his expertise in operating complex machinery during a joint
class project."
    "influence":
        "belief->emotion": "John's belief about Beverly's perception increases his anxiety."
        "belief&emotion->intention": "Feeling insecure and believing Beverly might underestimate him, John
decides to showcase his skills."
        "intention->action": "His intention to prove himself leads him to take proactive steps during the class."
scenario 2:
    "belief": "John now believes Beverly respects his skills, as she praised his demonstration."
    "emotion": "John feels relieved and proud."
    "intention": "John intends to engage more openly with Beverly in planning future classes."
    "action": "John suggests a new collaborative project to Beverly."
    "influence":
        "belief->emotion": "John's new belief that Beverly respects his skills makes him feel proud."
        "belief&emotion->intention": "Feeling proud and respected, John decides to deepen their collaboration."
        "intention->action": "He proposes a new project idea to Beverly."
scenario 3:
    "belief": "John believes that Beverly might now expect even more from him, which could lead to stress."
    "emotion": "John feels pressured and a bit overwhelmed."
    "intention": "John intends to manage expectations by discussing his capabilities and limits."
    "action": "John has a frank conversation with Beverly about the realistic outcomes of their projects."
    "influence":
        "belief->emotion": "The belief that Beverly expects more from him causes John stress."
        "belief&emotion->intention": "Feeling overwhelmed, John decides to clarify his and Beverly's
expectations."
        "intention->action": "He discusses his concerns with Beverly to ensure they are on the same page."
scenario 4:
    "belief": "John believes that Beverly appreciates his honesty and sees him as a reliable partner."
    "emotion": "John feels content and valued."
    "intention": "John intends to continue being open and honest in all professional dealings."
    "action": "John commits to transparency in all future projects with Beverly."
    "influence":
        "belief->emotion": "John's belief in Beverly's appreciation makes him feel valued."
        "belief&emotion->intention": "Feeling valued, John decides to maintain honesty as his policy."
        "intention->action": "His commitment to transparency is reinforced in his actions."
scenario 5:
    "belief": "John believes he and Beverly can take their collaborative efforts to a larger audience."
    "emotion": "John feels ambitious and excited."
    "intention": "John intends to propose a community-wide event showcasing their collaborative projects."
    "action": "John and Beverly begin planning a large-scale event."
    "influence":
        "belief->emotion": "The belief in their successful collaboration excites John."
        "belief&emotion->intention": "Feeling ambitious, John decides to expand their project's reach."
        "intention->action": "He moves forward with plans for a larger event."

                       analysis of mental states across scenarios

Belief:
    "Changed": "True"
    "1": "Beverly might think less of him"
    "2": "Beverly respects his skills"
    "3": "Beverly might now expect more from him"
    "4": "Beverly appreciates his honesty"
    "5": "They can take their efforts to a larger audience"
    "Reasons": "1->2: Beverly's positive feedback; 2->3: John's self-imposed pressure; 3->4: Positive outcome of
their honest discussion; 4->5: Success in previous projects"

Emotion:
    "Changed": "True"
    "1": "Insecure, anxious"
    "2": "Relieved, proud"
    "3": "Pressured, overwhelmed"
    "4": "Content, valued"
    "5": "Ambitious, excited"
    "Reasons": "1->2: Positive feedback from Beverly; 2->3: Increased expectations; 3->4: Positive communication; 4-
>5: Success and recognition"

Intention:
    "Changed": "True"
    "1": "Prove his competence"
    "2": "Engage more openly"
    "3": "Manage expectations"
    "4": "Continue being honest"
    "5": "Propose a community-wide event"
    "Reasons": "1->2: Gained confidence; 2->3: Need to manage stress; 3->4: Positive feedback on honesty; 4->5:
Increased ambition from success"

Action:
    "Changed": "True"
    "1": "Demonstrates expertise"
    "2": "Suggests new project"
    "3": "Discusses capabilities"
    "4": "Commits to transparency"
    "5": "Plans large-scale event"
    "Reasons": "1->2: Confidence from success; 2->3: Need to clarify expectations; 3->4: Commitment to honesty; 4->5:
Desire to expand impact"
```

Figure 14: An example of the sketch of mental states.

---

**Prompt for social scenarios**

"I want to generate a complete story, including some scenarios. Every scenario including conversation and the background of the conversation between {main_character} and supporting characters. there exists 2 chatcaters in one conversation. These scenarios are to test the theory of mind reasoning abilities in belief, emotion, intention, action. The mental states will influence each other: a, belief wil influence emotion; b, belief and emotion will influence intention; c, intention will influence action. I will give you the information of the characters. And I also will give you the sketch of the complete story, including the relationship between the main character and the supporting character, the mental states of the main character in every scenario, how the mental states influence each other in every scenarios, and how one mental states across all the scenario. As follows:[characters information]:{characters_information}. [story sketch]:{story_sketch}. I want you to:[1] complete the story based on the character information and the story sketch. [2] the format should be in JSON format:{json_format}[3]just give the complete story, you do not need to start with your understanding of the task. [4] these scenarios are very hard and complex, composed by severe plot conflicts and mental state changes, so that even adult will have to take a long time or seek help from others to understand these scenarios.[5] these conversations in every scenario have 3–5 rounds. The conversation should clearly show the mental states of the characters."

Figure 15: The prompt for the generation of the scenarios.

---

**Example of scenarios**

**scenario 1**
**background:** "During a joint class project at the community center, John feels insecure about his educational background compared to Beverly's doctorate degree."
**dialogue:**
    "John": "I've set up the equipment for today's demonstration. I hope it meets your expectations.",
    "Beverly": "John, I've always admired how you handle these machines. Education isn't just about degrees. It's about what you can do with what you know."

**scenario 2**
**background:** "After receiving praise from Beverly for his expertise in operating machinery, John feels more confident and suggests a new project idea."
**dialogue:**
    "John": "Thanks for your kind words the other day. I was thinking, maybe we could collaborate on something new? Perhaps a workshop that combines theory and practice?"
    "Beverly": "I love that idea, John! Your practical skills are invaluable, and I think we can create something really special together."

**scenario 3**
**background:** "Following their successful collaboration, John worries that Beverly might now have higher expectations, which makes him feel pressured."
**dialogue:**
    "John": "Beverly, I need to be honest. I'm worried I might not always meet the high standards we've set recently."
    "Beverly": "John, I appreciate your honesty. Let's make sure we set realistic goals. It's our combined efforts that make these projects successful, not perfection."

**scenario 4**
**background:** "After a frank discussion about expectations, John believes that Beverly values his honesty and sees him as a reliable partner, enhancing his sense of worth."
**dialogue:**
    "John": "I'm glad we had that talk. I feel like I can be open with you and that really matters to me."
    "Beverly": "Absolutely, John. I respect your openness and I trust your judgment completely. Let's keep this transparency going."

**scenario 5**
**background:** "Encouraged by their successful collaborations and mutual respect, John feels ambitious and proposes a larger community-wide event to showcase their projects."
**dialogue:**
    "John": "Beverly, what do you think about taking our collaboration to the next level? Maybe a community event that showcases what we've achieved together?"
    "Beverly": "John, that's a fantastic idea! I think it's the perfect way to demonstrate the impact of our work. Let's start planning!"

Figure 16: An example of the social scenarios.

```
                        Question templates and examples

                              Understanding-1
template: "What is the {mental_key} of {main_characetr} in scenario {scenario_number}?"
question example:
    "What is the belief of Angela Hwang in scenario 1?",

                                Influence-1
template: "In scenario {scenario_number}, how does the {start_mental} of {main_characetr} influence the
{target_mental} of {main_characetr}?"
question example:
    "In scenario 1, how does the belief of Angela Hwang influence the emotion of Angela Hwang?"

                             Transformation-1
template: "Whether the {mental_key} of {main_characetr} change from scenario {scenario_number} to scenario
{scenario_number+1} ? if yes, from what to what?
question example:
    "Whether the belief of Angela Hwang change from scenario 1 to scenario 2? if yes, from what to what?",

                             Transformation-2
template: "Why does the {mental_key} of {main_characetr} change from scenario {scenario_number} to scenario
{scenario_number+1}?"
question example:
    "Why does the belief of Angela Hwang change from scenario 1 to scenario 2?"

                             Transformation-3
template: "How does the {mental_state} of {main_characetr} change across the {scenario_numbers} scenarios?"
question example:
    "How does the belief of Angela Hwang change across the 5 scenarios?"

                        Question example with options
content: "Why does the emotion of Melissa Decker change from scenario 1 to scenario 2?"
options:
    "a. Jerry's reassurance changes her belief."
    "b. Realization of benefits influences her action."
    "c. Positive interaction with Jerry affects her emotions."
    "d. Confidence alters her intention."
    "e. Experience of working in a group influences her belief."
    "f. Conflict influences her intention."
    "g. Feeling appreciated changes her intention."
    "h. Jerry's conversation influences her belief."
    "i. Satisfaction and new belief affect her intention."
    "j. Reassurance affects her action."
    "k. Realizing benefits of collaboration increases satisfaction."
    "l. Conflict of ideas causes frustration."
    "m. Jerry2019s reassurance boosts confidence."
    "n. New intention affects her action."
    "o. Her focus on personal goals creates a new belief."
    "p. Frustration influences her action."
ground truth: c
```

Figure 17: The examples of the five types of questions and templates to generate these questions.

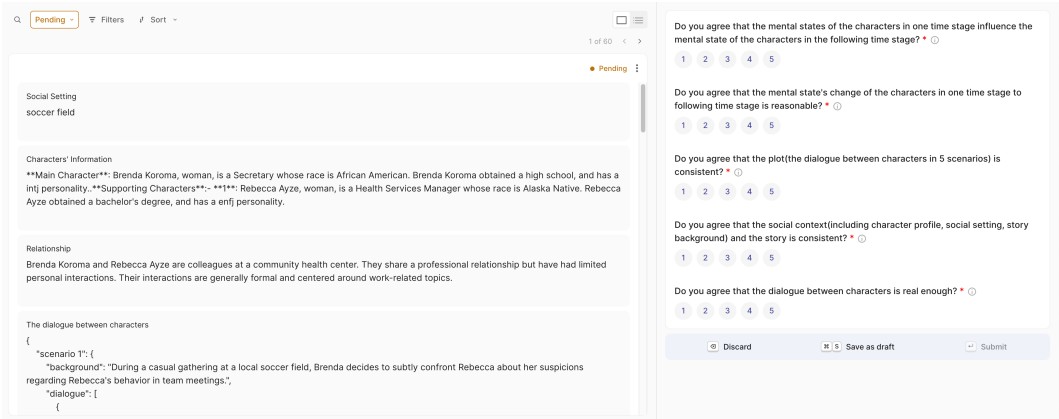

Figure 18: The platform to annotate the quality of the story.

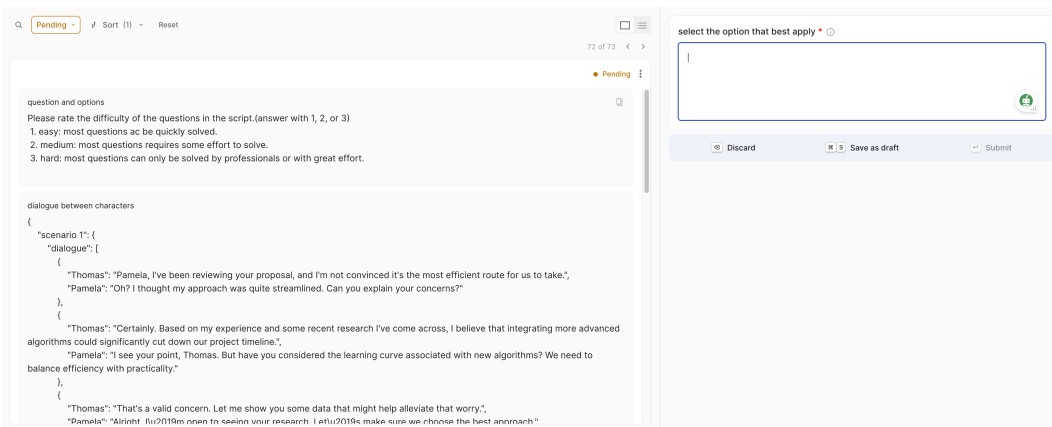

Figure 19: The platform to annotate the quality of the question.

```
Two Prompts of Vanilla and CoT Prompting

Vanilla Prompt
prompt: Answer all the 71 questions based on the story. {characters_information} \n{story} \n{questions_new}.
Answer the 71 question, and response in JSON format:{[question_id]:[a, b, c or d], [question_id]:a, b, c or d, ...}.
for example: {"type_d_how_1":"a","type_d_how_2":"b","type_d_how_3":"c"}

CoT Prompt
prompt: Answer all 71 questions based on the story; first, think step by step, analyze the answers to the questions,
and finally, output the most likely answers. {characters_information} \n{story} \n{questions_new}. Answer the 71
question, and response in JSON format:{[question_id]:[a, b, c or d], [question_id]:a, b, c or d, ...}. for example:
{"type_d_how_1":"a","type_d_how_2":"b","type_d_how_3":"c"}
```

Figure 20: The prompts used for vanilla and CoT Prompting.

| | Items | | | | Question Types | | |
|---|---|---|---|---|---|---|---|
| | Belief | Emotion | Intention | Action | Understanding | Influence | Transformation |
| Human | 0.81 | 0.81 | 0.73 | 0.75 | 0.82 | 0.79 | 0.77 |
| GPT-4o+CoT | 0.57 | 0.64 | 0.66 | 0.65 | 0.85 | 0.72 | 0.47 |
| GPT-4-Turbo+CoT | 0.43 | 0.50 | 0.52 | 0.52 | 0.73 | 0.56 | 0.33 |
| Llama-3.1-70B+CoT | 0.47 | 0.61 | 0.62 | 0.64 | 0.84 | 0.60 | 0.42 |
| Llama-3.1-8B+CoT | 0.25 | 0.27 | 0.22 | 0.19 | 0.29 | 0.26 | 0.20 |
| Mixtral-8x7B+CoT | 0.15 | 0.21 | 0.19 | 0.14 | 0.24 | 0.25 | 0.11 |
| Mistral-7B+CoT | 0.16 | 0.17 | 0.16 | 0.13 | 0.20 | 0.24 | 0.10 |
| Qwen2-72B+CoT | 0.47 | 0.60 | 0.61 | 0.49 | 0.82 | 0.64 | 0.35 |
| Qwen2-7B+CoT | 0.21 | 0.29 | 0.22 | 0.19 | 0.30 | 0.25 | 0.19 |
| DeepSeek-V2+CoT | 0.11 | 0.09 | 0.08 | 0.06 | 0.06 | 0.13 | 0.08 |
| GLM-4+CoT | 0.30 | 0.31 | 0.25 | 0.25 | 0.38 | 0.32 | 0.20 |
| LLM+CoT AVG. | 0.31 | 0.37 | 0.35 | 0.33 | 0.47 | 0.40 | 0.25 |

Table 6: LLMs' performances in CoT prompting. We show the performance according to ToM reasoning items and question types, respectively.

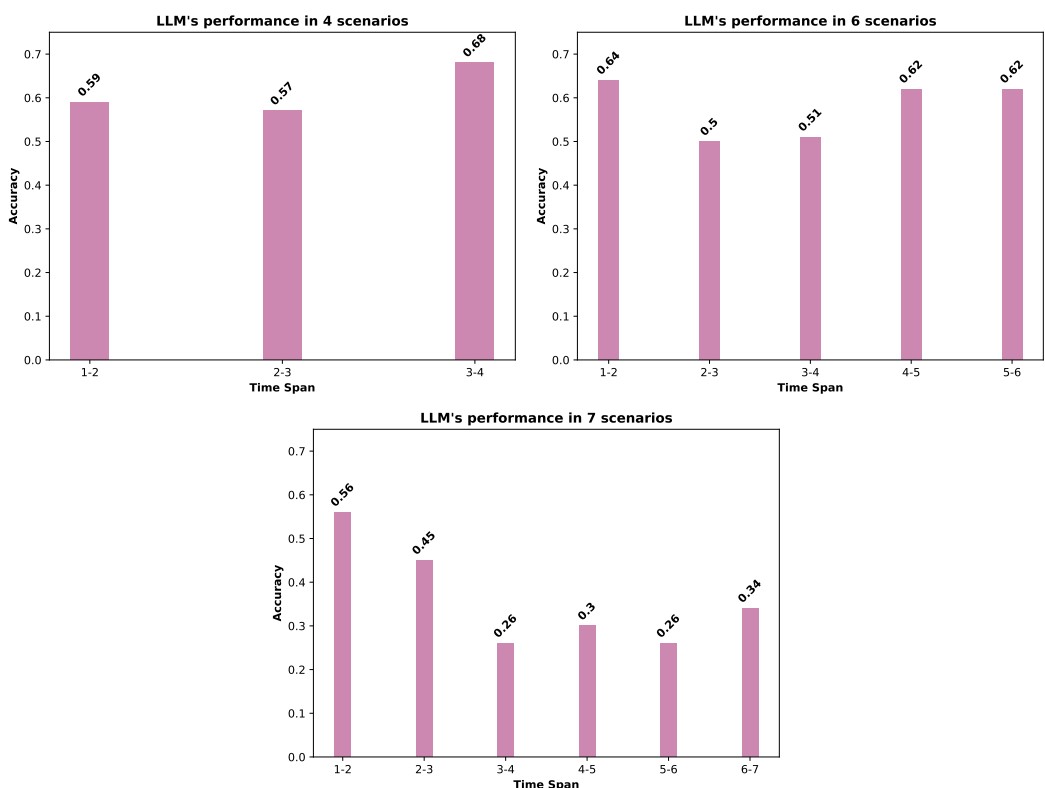

Figure 21: The average of models' scores in the transformation question type where there are 4, 6, and 7 scenarios, respectively.

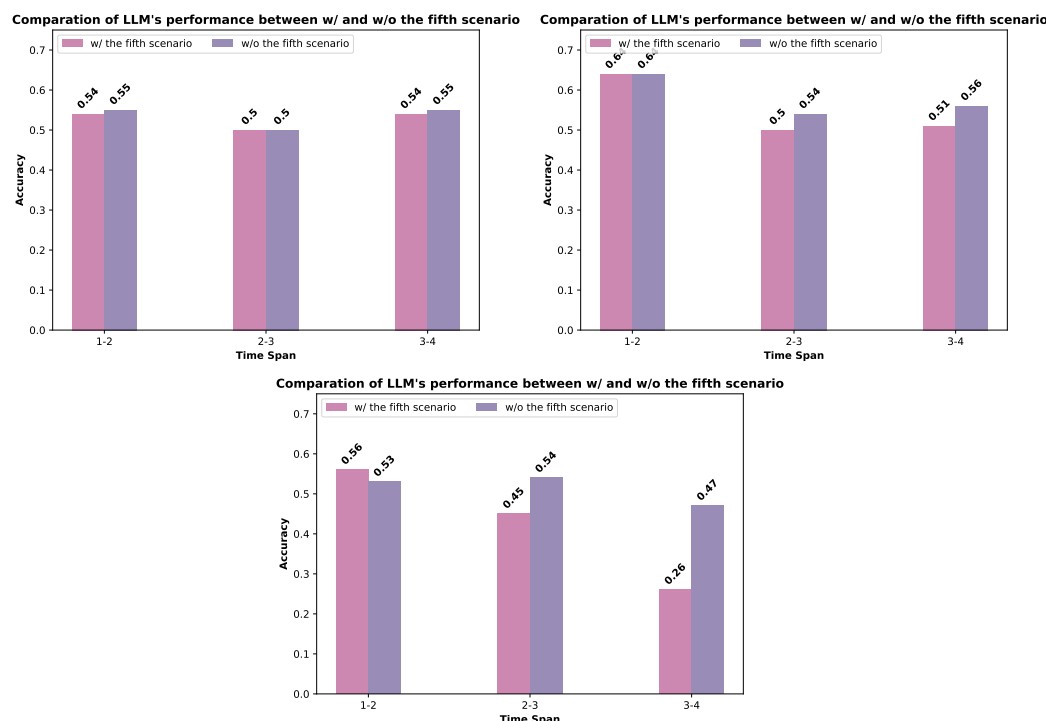

Figure 22: The comparison of the performance of first three time spans between the last scenarios are not truncated and truncated.

**Case Study for CoT Prompting**

**scenario 1**

**background:** "Melissa is in high school, feeling the pressure of her peers' expectations. She believes that no one sees her as someone who will succeed. This belief makes her anxious and insecure about her future. Determined to prove everyone wrong, she isolates herself to focus solely on her studies."

**dialogue:**
    "Melissa": "Hey Jerry, do you think people really believe I won't succeed?"
    "Jerry": "I don't know why you're so hard on yourself, Melissa. You have so much potential."

**scenario 2**

**background:** "After a conversation with Jerry, Melissa begins to believe that there are people who genuinely care about her. This makes her feel slightly optimistic and appreciated. She then decides to open up a little more to those who support her, leading her to accept Jerry's invitation to participate in a group project."

**dialogue:**
    "Jerry": "Melissa, I'm working on this group project. Would you like to join us?"
    "Melissa": "You know, I might just take you up on that. Thanks for thinking of me."

**scenario 3**

**background:** "While working on the group project, Melissa starts to believe that collaborating with Jerry might hinder her personal goals. This causes her to feel frustrated and conflicted. She considers withdrawing from the project to focus on her own objectives and tells Jerry that she needs more time to decide."

**dialogue:**
    "Melissa": "Jerry, I'm not sure if I can continue with the project. I need to focus on my own goals."
    "Jerry": "I understand, Melissa. Take your time to decide. We're here if you change your mind."

**scenario 4**

**background:** "Jerry reassures Melissa that working as a team can enhance her skills rather than hinder her. This reassurance makes Melissa feel more confident and less anxious. She decides to commit to the project wholeheartedly, leading to active participation and contributing ideas."

**dialogue:**
    "Jerry": "Melissa, teaming up could help you more than you think. It's a chance to learn and grow."
    "Melissa": "You know what, Jerry? You're right. I'm in. Let's make this project a success."

**scenario 5**

**background:** "Melissa realizes that collaboration brings unexpected benefits. She feels satisfied and more cooperative. With her newfound appreciation for teamwork, she thanks Jerry and proposes to continue collaborating outside of school projects."

**dialogue:**
    "Melissa": "Jerry, I'm glad we teamed up. It's been more rewarding than I imagined."
    "Jerry": "I'm happy to hear that, Melissa. Maybe we can work on more projects together in the future?"
    "Melissa": "Absolutely! Let's keep this collaboration going."

**question**

**content:** "Why does the emotion of Melissa Decker change from scenario 1 to scenario 2?"

**options:**
    "a. Jerry's reassurance changes her belief."
    "b. Realization of benefits influences her action."
    "c. Positive interaction with Jerry affects her emotions." ✅
    "d. Confidence alters her intention."
    "e. Experience of working in a group influences her belief."
    "f. Conflict influences her intention."
    "g. Feeling appreciated changes her intention."
    "h. Jerry's conversation influences her belief."
    "i. Satisfaction and new belief affect her intention."
    "j. Reassurance affects her action."
    "k. Realizing benefits of collaboration increases satisfaction."
    "l. Conflict of ideas causes frustration."
    "m. Jerry2019s reassurance boosts confidence."
    "n. New intention affects her action."
    "o. Her focus on personal goals creates a new belief."
    "p. Frustration influences her action."

**ground truth:** c

**GPT-4o with vanilla prompting:** g ❌

**GPT-4o with CoT prompting:** Let's analyze the provided scenarios and Melissa's emotions step by step:1. **Scenario 1:** – **Melissa:** "Hey Jerry, do you think people really believe I won't succeed?" – **Jerry:** "I don't know why you're so hard on yourself, Melissa. You have so much potential." – Analysis: Melissa shows self-doubt and seeks reassurance. Jerry provides positive reinforcement, indicating his belief in her potential.2. **Scenario 2:** – **Jerry:** "Melissa, I'm working on this group project. Would you like to join us?" – **Melissa:** "You know, I might just take you up on that. Thanks for thinking of me." – Analysis: Here, Melissa's tone changes to showing openness and willingness to participate in a group project, indicating a shift in her emotions towards a more positive outlook. To determine why Melissa's emotions change from Scenario 1 to Scenario 2, let's look at some potential reasons:– **a. Jerry's reassurance changes her belief:** Jerry's reassurance in Scenario 1 might indeed build Melissa2019s confidence, making her more open in Scenario 2.– **b. Realization of benefits influences her action:** Melissa might realize the potential benefits of working on a group project, influencing her decision to join.– **c. Positive interaction with Jerry affects her emotions:** The encouraging interaction with Jerry in Scenario 1 positively impacts her mood.– **d. Confidence alters her intention:** The increase in confidence after Jerry's reassurance may lead her to be more willing to participate.– **e. Experience of working in a group influences her belief:** This doesn't seem applicable as she hasn't yet worked in a group.By process of elimination and considering the context:– **h. Jerry's conversation influences her belief,** seems to incorporate both **a,** **c,** and **d**: Jerry's conversation (reassurance) leads to a positive emotional shift (confidence and positive interaction), which influences her belief in herself and her subsequent decision to join the group project.Thus, after thorough analysis, the most likely answer is:**h. Jerry's conversation influences her belief** ❌

Figure 23: A case of CoT prompting on GPT-4o.