# OpenReview forum: "TOMVALLEY: EVALUATING THE THEORY OF MIND REASONING OF LLMS IN REALISTIC SOCIAL CONTEXT"
_ICLR.cc/2025/Conference — ICLR 2025 Conference Withdrawn Submission_

### Official Review · Reviewer_UvAb · 2024-10-17

**Soundness:** 3
**Presentation:** 2
**Contribution:** 3
**Rating:** 5
**Confidence:** 4

**Summary:**

The paper introduces TOMVALLEY, a new benchmark designed to evaluate LLM's theory of mind (ToM) capabilities in realistic social contexts. TOMVALLEY provides 1,100 diverse social contexts and 78,100 questions about characters' mental states, constructed through a systematic framework that includes social background determination, mental state sketching, social scenario design, and rule-based question generation. The authors manually verified the quality of the benchmark to ensure its reliability. They evaluated ten popular LLMs using TOMVALLEY and found that their best performance lagged behind human levels by 11%. The results indicate that LLMs struggle to interpret changes in mental states across scenarios, providing insights into the current limitations of LLMs in understanding complex social interactions.

**Strengths:**

1. The authors present TOMVALLEY, a new benchmark that realistically mirrors social contexts. It includes a wide range of social settings, character profiles, and relationships between characters. This diversity is achieved through a systematic construction framework, making the benchmark both comprehensive and relevant to real-world scenarios.
2. The authors have manually verified the design of the benchmark to ensure its quality and reliability. They not only provide a human performance baseline for comparison but also conduct a thorough quality assessment.
3. The authors evaluate 10 LLMs on the benchmark, offering a lot of data to understand the reasoning of different LLMs.
4. The authors offer interesting analyses that shed new light on the area. For example, they explore how the presence or absence of character profiles affects model performance, reveal that LLMs Fail in the Middle Scenario, and demonstrate the models' limitations in handling compositional questions that require rigorous multi-hop reasoning to reach the answer.

**Weaknesses:**

1. There's a lack of comprehensive discussion of existing benchmarks. While the authors point out limitations in the previous theory of mind (ToM) benchmarks—such as focusing on static or independent mental states and lacking social settings—they do not adequately compare their work to existing studies. For example, benchmarks like MMToM-QA [1] and MuMA-ToM [2] explore interconnected mental states by asking about one mental state conditioned on another. Benchmarks like OpenToM [3] include narrative stories and character profiles, addressing the social context aspect. There are other benchmarks like EmoBench [4] and Infant Cognition Benchmark [5] that are missed in Table 2.

2. In Section 2.2, the paper introduces the concept of "process-level evaluation" but fails to provide a clear definition for it. This makes it difficult for readers to grasp what this term means and how it relates to the papers' objectives. The coherence of the paragraph can also be improved.

3. There's insufficient explanation of question types and categories. The benchmark includes items related to belief, emotion, intention, and action, and categorizes questions into types like understanding, influence, and transformation. However, these categories are only briefly mentioned without clear definitions or justification for their inclusion. Furthermore, a deeper analysis is needed to explain why LLMs' performance varies significantly across different question types, which could offer valuable insights into the models' strengths and weaknesses.

4. The authors mention that the benchmark includes compositional problems that require rigorous multi-hop reasoning to arrive at the correct answer. However, they do not clearly define what these problems entail or how they were constructed to necessitate such reasoning. Providing a detailed explanation and concrete examples would help readers understand their importance and how they challenge LLMs differently than simpler questions.

5. While the benchmark incorporates social backgrounds, character profiles, and relationships, and claims that humans effortlessly use this information in ToM reasoning, the authors don't clearly explain how this information affects the answers to the questions. It remains unclear whether these details are essential for arriving at the correct answers or if they simply add diversity to the questions. A better explanation is needed to illustrate how social context influences the reasoning process and why it is crucial for evaluating ToM capabilities in LLMs.

**Questions:**

1. While I appreciate the effort of testing 10 different LLMs, have you considered evaluating the ToM methods such as Sim-ToM [6], SymbolicToM [7] (ACL 2023 Outstanding Paper Award), and BIP-ALM [1] (ACL 2024 Outstanding Paper Award) on your benchmark? These approaches have demonstrated notable improvements in LLM reasoning and have "solved" some prior benchmarks.

2. Just out of curiosity, could you also test GPT-4o on your benchmark? GPT-4o has shown significantly enhanced performance on some previous benchmarks related to ToM tasks. Feel free to disregard this question if you are constrained by time or resources during the rebuttal, but perhaps testing a small portion of your benchmark with GPT-4o could provide useful insights.

3. Since you used GPT-4-Turbo to generate the dialogues and scenarios for your benchmark, could this influence the performance of the same or similar models when they are tested on TOMVALLEY? Might the similarity in writing style or content affect the evaluation results due to the models being tested on data generated by themselves?

4. Regarding the human evaluation of question quality, could you briefly clarify (in one sentence) how you ensure fairness and objectivity in this process? I’m somewhat concerned that if the "five graduate students" are from the same lab, it could introduce bias.

5. Small typo: Line 250: "toM" -> "ToM".

References:
1. MMToM-QA: Multimodal Theory of Mind Question Answering, Jin et al, 2024
2. MuMA-ToM: Multi-modal Multi-Agent Theory of Mind, Shi et al, 2024
3. OpenToM: A Comprehensive Benchmark for Evaluating Theory-of-Mind Reasoning Capabilities of Large Language Models, Xu et al, 2024
4. EmoBench: Evaluating the Emotional Intelligence of Large Language Models, Sabour et al, 2024
5. An Infant-Cognition Inspired Machine Benchmark for Identifying Agency, Affiliation, Belief, and Intention, Li et al, 2024
6. Think Twice: Perspective-Taking Improves Large Language Models' Theory-of-Mind Capabilities, Wilf et al, 2023
7. Minding Language Models' (Lack of) Theory of Mind: A Plug-and-Play Multi-Character Belief Tracker, Sclar et al, 2023

---

### Official Review · Reviewer_TrER · 2024-10-27

**Soundness:** 3
**Presentation:** 3
**Contribution:** 2
**Rating:** 3
**Confidence:** 5

**Summary:**

Overall, TOMVALLEY builds upon previous studies, notably synthesizing elements from OpenToM and BigToM. The dataset combines features such as plot, character profiles, relationships (inspired by OpenToM), and elements of social location and dynamic mental states (from BigToM). Like these works, TOMVALLEY also uses LLMs to generate richer scenes from provided information, using these scenes as labeled synthetic data for testing. While TOMVALLEY integrates these aspects, the approach and methodology align closely with earlier work, raising questions about its degree of novelty.

**Strengths:**

1. The authors have developed detailed character profiles that serve as the basis for generating dialogue, which adds depth and context to the interactions. Based on the appendix, TOMVALLEY covers a broader range of scenarios and settings than previous ToM datasets, making it a more comprehensive resource. The dialogue generation also appears carefully constructed, with prompts that result in good-quality conversations.

2. Unlike previous datasets that often include changes in objective factors, TOMVALLEY is fully focused on capturing and evaluating characters' mental states, with a stronger emphasis on emotions and psychological aspects. While OpenToM also explores character profiles, it primarily focuses on how these profiles influence characters' actions. In contrast, TOMVALLEY concentrates more deeply on mental states, making it a distinct contribution in this area compared to earlier works.

**Weaknesses:**

1. While ToM is an interesting and promising field, TOMVALLEY largely builds on prior work, offering incremental improvements rather than a completely new approach. The overall structure and evaluation strategy closely resemble past studies, making this contribution feel less innovative.
As I mentioned, the dataset combines features such as plot, character profiles, relationships (like OpenToM), and elements of social location and dynamic mental states (like BigToM). Similar to these works, TOMVALLEY also uses LLMs to generate richer scenes from provided structured information, using these scenes as labeled synthetic data for testing (where the generated scene serves as the input, and the structured information as the label). The evaluation method is also very similar to BigToM.
While following the choices of previous works is reasonable, the authors overstate their contribution by claiming: “previous works (1) most evaluations focus on a static mental state after several social scenarios, ignoring the changes in mental states across different scenarios; (2) they mainly consider independent mental states, whereas different kinds of mental states (beliefs, intentions, and emotions) and actions can influence one another in real life; (3) social settings and character profiles are absent in their evaluations, even though humans can effortlessly obtain and utilize this information in ToM reasoning processes."

2. Though TOMVALLEY has created a large dataset, its reliance on synthetic data introduces inevitable noise. The authors evaluated answers across five criteria, with around 90% of responses meeting each criterion individually. However, it would be helpful to know how many answers meet all five criteria simultaneously, as this would provide a more holistic view of response quality. (See questions below)

3. Additionally, the design of certain questions can be ambiguous. For instance, in a dialogue context, a question like “What is Angela Hwang's belief?” could be unclear without explicit contextual information, making it difficult for users to discern the exact belief being asked about.  (See questions below)

4. The analysis of model limitations also feels somewhat surface-level. It suggests that models struggle due to issues like “lost in the middle” phenomena and a lack of necessary ToM reasoning abilities. While these are valid points, they echo discussions already present in prior ToM research referenced by TOMVALLEY. Adding fresh insights or diving deeper into unique findings from this specific dataset could make the analysis more impactful and distinctive.

**Questions:**

1. While human annotators assessed the quality of generated social scenarios and sampled questions, there doesn’t seem to be an evaluation criterion to check if the answers align with the generated scenarios. Could the authors elaborate on why this alignment check wasn’t included?

2. The paper provides evaluations for each of the five criteria individually. However, it would be valuable to understand how many scenarios and question-answer pairs meet all five criteria simultaneously. This holistic assessment could give a more comprehensive measure of quality. Do the authors have data on the percentage of scenes and Q&A pairs that satisfy all dimensions?

3. Some questions, like “What is Angela Hwang’s belief?”, seem ambiguous without clear context. This could impact the clarity and effectiveness of the dataset, but change those question at this stage would be expensive. How do the authors plan to address this issue?

---

### Official Review · Reviewer_sGhK · 2024-11-02

**Soundness:** 2
**Presentation:** 2
**Contribution:** 2
**Rating:** 3
**Confidence:** 4

**Summary:**

The paper proposes a new theory of mind benchmark for LLMs called ToM Valley. The benchmark aims to address the gap of ToM being tested in limited, constrained scenarios by procedurally generating diverse scenarios, personas and conversations. The authors query for 4 types of inferences, belief, emotion, intention, and action with 3 types of questions. The authors validate their stimuli by asking grad students to rate them. The authors test 9 LLMs with 0-shot and 0-shot cot prompting. They find that the leading LLM trails human performance on the task. Through various ablations, the authors point to LLMs being bad at paying attention to information in the middle of the context, llms struggling at multi-hop compositional reasoning as issues.

**Strengths:**

- The paper is well motivated and addresses a gap between abstract tests of ToM and realistic contexts in which LLMs are used.
- I liked the diversity of scenarios present in the evaluation
- The authors have a diverse set of LLMs that they test.
- I liked the analysis of including and excluding personas, and also the failing in the middle analysis!

**Weaknesses:**

- Abstract:
    - Can be more specific about what exactly the authors are testing with the benchmark.
- In the introduction, it is not clear what exactly the questions are testing, this could be made much clearer. The Introduction and the abstract are very vague. They could be much more specific: 1) What are you testing 2) How exactly is the data generated 3) How do you validate the data 4) How do llms perform? Maybe using figure 1 as a reference would be useful
- Section 3: I am not convinced that the authors address the problem of circularity. The evaluation of a conversation being coherent to the mental states present and the dynamics of emotions in a social scenario is quite involved. The model generating these scenarios should then have a rich model of human mental states (this has not been validated in previous work).
    - Step 1: The way the profiles and names are collected could be moved from the appendix to the main text. Why are occupations pooled by gender? These could reinforce stereotypes.
    - Step 2: How are initial and final mental states decided upon?
    - Step 3: 1) How do you ensure that the dialogues generated by an LLM are faithful to the mental states? 2) How do you ensure that the conversation obeys the dynamics of the mental states (decided in step 2) and that these correspond to true dynamics of human mental states?  While the authors collect human judgements for these, are coherence, authenticity and dynamism mostly dependent on an LLM being good at being good enough at understanding social contexts?
    - How are the answer options for the questions generated? How is the correct answer picked? How do you ensure that all the other options are incorrect? For example, “how does … change across all social scenarios?” seems like a broad open-ended question.
    - In general, how do you ensure that there is no problem of circularity? That the capability being evaluated is not being used while generating the data?
- Confidence intervals are missing throughout the paper
- Why are myers-briggs personality traits used? These lack scientific validity and soundness. Why aren't alternatives like the big 5 / OCEAN used?
- An actual example like in fig16 would make the paper much clearer.
- It is not clear, what precise capabilities the benchmark is testing and why the construct is valid.
- The limitations are buried in the appendix, and should be addressed in the main draft.
- How many annotations per question do you collect? For both human experiments?

Minor:

It would be nice to see other closed source models like gemini, claude! But this is not necessary.

In the title “realistic social context” → “realistic social contexts”

Line 250: toM → ToM

Line 301: five 5-Likert → the five questions on a 5 point Likert scale

Line 789 qomplexity → complexity

**Questions:**

Please see weaknesses.

---

### Official Review · Reviewer_hdht · 2024-11-03

**Soundness:** 3
**Presentation:** 3
**Contribution:** 2
**Rating:** 5
**Confidence:** 4

**Summary:**

This paper presents a novel benchmark for evaluating the Theory of Mind (ToM) capabilities of large language models (LLMs). It conducts a comprehensive evaluation of ten popular LLMs. The experimental section provides an in-depth analysis and discussion on various aspects, including the compositional reasoning abilities of LLMs in ToM and comparisons with human performance.

**Strengths:**

This paper presents a novel benchmark for evaluating the ToM capabilities of LLMs, which takes into account the mental states of characters within real social contexts and the dependencies among fine-grained mental states.

The types of questions are intriguing, covering understanding, influence, and transformation.

The experiments include an in-depth analysis of combinatorial reasoning involved in ToM and middle scenarios.

**Weaknesses:**

The design of character profiles has been mentioned in the OpenToM literature [1]. The five constructed scenarios, when connected, are quite similar to a long conversation, as in FanToM [2], where ToM capabilities of LLMs are also assessed in a dialog format, and the characters' belief states evolve as the conversation progresses. (I acknowledge that this paper's settings are more grounded in real social contexts, distinguishing it from FanToM.)

For influence-type questions, such as Scenario1->Scenario2, would the model's performance significantly improve if only these two scenarios were provided? Does context length greatly impact the model's performance?

In scenarios resembling real social contexts, interactive evaluation may be a better approach than directly feeding conversations into LLMs.

[1] Xu H, Zhao R, Zhu L, et al. OpenToM: A Comprehensive Benchmark for Evaluating Theory-of-Mind Reasoning Capabilities of Large Language Models[J]. arXiv preprint arXiv:2402.06044, 2024.

[2] Kim H, Sclar M, Zhou X, et al. FANToM: A benchmark for stress-testing machine theory of mind in interactions[J]. EMNLP 2023.

**Questions:**

How are the options specifically designed, and how is the model's performance calculated?

In Table 3, belief, emotion, and intention all include influence-type questions. Figure 1 provides an example of how belief can influence emotion. So, what kind of relationships are relied upon in the design of influence questions for emotion and intention?

---

### Official Review · Reviewer_JF8n · 2024-11-04

**Soundness:** 1
**Presentation:** 1
**Contribution:** 1
**Rating:** 3
**Confidence:** 4

**Summary:**

The paper introduces TOMVALLEY, a benchmark designed to evaluate LLMs' Theory of Mind (ToM) reasoning. Authors claim that TOMVALLEY addresses limitations of existing benchmarks by incorporating dynamic and diverse mental states across multiple scenarios and detailed character profiles within specific social context. This benchmark includes 1100 social contexts with 78,100 questions about mental states and evaluates LLMs on their ability to reason about beliefs, emotions, intentions, and actions over time. They run evaluation by feeding 71 questions within a single prompt and measure their performance by parsing out the 71 answers from the output response. They use this procedure even in the CoT setup. Results show that models are lagging behind human performance by 11% and CoT does not help.

**Strengths:**

Incorporating dynamic mental states and multiple dimensions is essential for measuring holistic ToM reasoning.

**Weaknesses:**

- The primary weakness of this paper is that all evaluations are conducted by presenting 71 questions within a single prompt (Figure 20). This setup, which is also used for CoT evaluation with claims that CoT does not enhance performance, introduces numerous confounding variables, making it unsuitable for benchmarking performance accurately. For example, results from the questions in the beginning will definitely impact later responses. Moreover, CoT is not suited for coming up with batch of answers. This is probably why the CoT does not show performance improvement. Since all of the results and analyses are based on this setup, I find it difficult to correctly interpret them. I strongly encourage the authors to rerun their experiments on each question independently.
- The benchmark is entirely generated by an LLM, but human validation was conducted on only 5% of the dataset—a very limited proportion. Since LLMs are known to struggle with theory of mind tasks, human validation is strongly recommended, especially given the complexity of the benchmark design, which involves multiple steps in the generation pipeline. I would suggest running at least 33% of the data. Ideally, I would encourage 100% of the data.

**Questions:**

- Some of the large figures could have been better if they were tables with a few rows. Why is Figure 3 a figure?
- “Meanwhile, ToM reasoning in dialogues has seldom been investigated in previous works.” → There are many existing works, such as
    - https://aclanthology.org/2023.emnlp-main.890/
    - https://aclanthology.org/2024.sigdial-1.63/
    - https://aclanthology.org/2024.acl-short.26/
    - I encourage the authors to compare their benchmark with these as well.
- I spotted many grammar issues, please fix them in the updated draft.

---

> ### Author Response · Authors · 2024-11-17
> **Thanks for you suggestion!**
>
> Thanks for your valuable suggestion!
>
> * “Meanwhile, ToM reasoning in dialogues has seldom been investigated in previous works.”
>
> Given a large amount of tom-related research, we contend that this description does not introduce significant ambiguity. We will take your advice and compare with these works.

---

### Note · Authors · 2024-11-27

**Comment:**

Thanks for your suggestions and questions！

**Withdrawal Confirmation:**

I have read and agree with the venue's withdrawal policy on behalf of myself and my co-authors.